# INDIVIDUAL PRIVACY ACCOUNTING WITH GAUSSIAN DIFFERENTIAL PRIVACY

**Antti Koskela**
Nokia Bell Labs
University of Helsinki
antti.h.koskela@nokia-bell-labs.com

**Marlon Tobaben**
University of Helsinki
marlon.tobaben@helsinki.fi

**Antti Honkela**
University of Helsinki
antti.honkela@helsinki.fi

## ABSTRACT

Individual privacy accounting enables bounding differential privacy (DP) loss individually for each participant involved in the analysis. This can be informative as often the individual privacy losses are considerably smaller than those indicated by the DP bounds that are based on considering worst-case bounds at each data access. In order to account for the individual privacy losses in a principled manner, we need a privacy accountant for adaptive compositions of randomised mechanisms, where the loss incurred at a given data access is allowed to be smaller than the worst-case loss. This kind of analysis has been carried out for the Rényi differential privacy by Feldman and Zrnic (12), however not yet for the so called optimal privacy accountants. We make first steps in this direction by providing a careful analysis using the Gaussian differential privacy which gives optimal bounds for the Gaussian mechanism, one of the most versatile DP mechanisms. This approach is based on determining a certain supermartingale for the hockey-stick divergence and on extending the Rényi divergence-based fully adaptive composition results by Feldman and Zrnic (12). We also consider measuring the individual $(\varepsilon, \delta)$-privacy losses using the so called privacy loss distributions. With the help of the Blackwell theorem, we can then make use of the results of Feldman and Zrnic (12) to construct an approximative individual $(\varepsilon, \delta)$-accountant.

## 1 INTRODUCTION

Differential privacy (DP) (8) provides means to accurately bound the compound privacy loss of multiple accesses to a database. Common differential privacy composition accounting techniques such as Rényi differential privacy (RDP) based techniques (23; 33; 38; 24) or so called optimal accounting techniques (19; 15; 37) require that the privacy parameters of all algorithms are fixed beforehand. Rogers et al. (28) were the first to analyse fully adaptive compositions, wherein the mechanisms are allowed to be selected adaptively. Rogers et al. (28) introduced two objects for measuring privacy in fully adaptive compositions: privacy filters, which halt the algorithms when a given budget is exceeded, and privacy odometers, which output bounds on the privacy loss incurred so far. Whitehouse et al. (34) have tightened these composition bounds using filters to match the tightness of the so called advanced composition theorem (9). Feldman and Zrnic (12) obtain similar $(\varepsilon, \delta)$-asymptotics via RDP analysis.

In their analysis using RDP, Feldman and Zrnic (12) consider individual filters, where the algorithm stops releasing information about the data elements that have exceeded a pre-defined RDP budget. This kind of individual analysis has not yet been considered for the optimal privacy accountants. We make first steps in this direction by providing a fully adaptive individual DP analysis using the Gaussian differential privacy (7). Our analysis leads to tight bounds for the Gaussian mechanism and it is based on determining a certain supermartingale for the hockey-stick divergence and on using similar proof techniques as in the RDP-based fully adaptive composition results of Feldman

and Zrnic (12). We note that the idea of individual accounting of privacy losses has been previously considered in various forms by, e.g., Ghosh and Roth (13); Ebadi et al. (10); Wang (32); Cummings and Durfee (6); Ligett et al. (22); Redberg and Wang (27).

We also consider measuring the individual $(\varepsilon, \delta)$-privacy losses using the so called privacy loss distributions (PLDs). Using the Blackwell theorem, we can in this case rely on the results of (12) to construct an approximative $(\varepsilon, \delta)$-accountant that often leads to smaller individual $\varepsilon$-values than commonly used RDP accountants. For this accountant, evaluating the individual DP-parameters using the existing methods requires computing FFT at each step of the adaptive analysis. We speed up this computation by placing the individual DP hyperparameters into well-chosen buckets, and by using pre-computed Fourier transforms. Moreover, by using the Plancherel theorem, we obtain a further speed-up.

## 1.1 OUR CONTRIBUTIONS

Our main contributions are the following:

- We show how to analyse fully adaptive compositions of DP mechanisms using the Gaussian differential privacy. Our results give tight $(\varepsilon, \delta)$-bounds for compositions of Gaussian mechanisms and are the first results with tight bounds for fully adaptive compositions.

- Using the concept of dominating pairs of distributions and by utilising the Blackwell theorem, we propose an approximative individual $(\varepsilon, \delta)$-accountant that in several cases leads to smaller individual $\varepsilon$-bounds than the individual RDP analysis.

- We propose efficient numerical techniques to compute individual privacy parameters using privacy loss distributions (PLDs) and the FFT algorithm. We show that individual $\varepsilon$-values can be accurately approximated in $\mathcal{O}(n)$-time, where $n$ is the number of discretisation points for the PLDs. Due to the lack of space this is described in Appendix D.

- We give experimental results that illustrate the benefits of replacing the RDP analysis with GDP accounting or with FFT based numerical accounting techniques. As an observation of indepedent interest, we notice that individual filtering leads to a disparate loss of accuracies among subgroups when training a neural network using DP gradient descent.

## 2 BACKGROUND

### 2.1 DIFFERENTIAL PRIVACY

We first shortly review the required definitions and results for our analysis. For more detailed discussion, see e.g. (7) and (37).

An input dataset containing $N$ data points is denoted as $X = (x_1, \ldots, x_N) \in \mathcal{X}^N$, where $x_i \in \mathcal{X}$, $1 \leq i \leq N$. We say $X$ and $X'$ are neighbours if we get one by adding or removing one element in the other (denoted $X \sim X'$). To this end, similarly to Feldman and Zrnic (12), we also denote $X^{-i}$ the dataset obtained by removing element $x_i$ from $X$, i.e.

$$X^{-i} = (x_1, \ldots, x_{i-1}, x_{i+1}, \ldots, x_N).$$

A mechanism $\mathcal{M}$ is $(\varepsilon, \delta)$-DP if its outputs are $(\varepsilon, \delta)$-indistinguishable for neighbouring datasets.

**Definition 1.** *Let $\varepsilon \geq 0$ and $\delta \in [0, 1]$. Mechanism $\mathcal{M} : \mathcal{X}^n \to \mathcal{O}$ is $(\varepsilon, \delta)$-DP if for every pair of neighbouring datasets $X, X'$, every measurable set $E \subset \mathcal{O}$,*

$$\mathbb{P}(\mathcal{M}(X) \in E) \leq e^{\varepsilon} \mathbb{P}(\mathcal{M}(X') \in E) + \delta.$$

*We call $\mathcal{M}$ tightly $(\varepsilon, \delta)$-DP, if there does not exist $\delta' < \delta$ such that $\mathcal{M}$ is $(\varepsilon, \delta')$-DP.*

The $(\varepsilon, \delta)$-DP bounds can also be characterised using the Hockey-stick divergence. For $\alpha > 0$ the hockey-stick divergence $H_\alpha$ from a distribution $P$ to a distribution $Q$ is defined as

$$H_\alpha(P||Q) = \int [P(t) - \alpha \cdot Q(t)]_+ \ dt,$$

where for $x \in \mathbb{R}$, $[x]_+ = \max\{0, x\}$. Tight $(\varepsilon, \delta)$-values for a given mechanism can be obtained using the hockey-stick-divergence:

**Lemma 2** (Zhu et al. 37). *For a given $\varepsilon \geq 0$, tight $\delta(\varepsilon)$ is given by the expression*

$$\delta(\varepsilon) = \max_{X \sim X'} H_{\mathrm{e}^\varepsilon}(\mathcal{M}(X) \| \mathcal{M}(X')).$$

Thus, if we can bound the divergence $H_{\mathrm{e}^\varepsilon}(\mathcal{M}(X) \| \mathcal{M}(X'))$ accurately, we also obtain accurate $\delta(\varepsilon)$-bounds. To this end we consider so called dominating pairs of distributions:

**Definition 3** (Zhu et al. 37). *A pair of distributions $(P, Q)$ is a* dominating pair *of distributions for mechanism $\mathcal{M}(X)$ if for all neighbouring datasets $X$ and $X'$ and for all $\alpha > 0$,*

$$H_\alpha(\mathcal{M}(X) \| \mathcal{M}(X')) \leq H_\alpha(P \| Q).$$

*If the equality holds for all $\alpha$ for some $X, X'$, then $(P, Q)$ is tightly dominating.*

Dominating pairs of distributions also give upper bounds for adaptive compositions:

**Theorem 4** (Zhu et al. 37). *If $(P, Q)$ dominates $\mathcal{M}$ and $(P', Q')$ dominates $\mathcal{M}'$, then $(P \times P', Q \times Q')$ dominates the adaptive composition $\mathcal{M} \circ \mathcal{M}'$.*

To convert the hockey-stick divergence from $P \times P'$ to $Q \times Q'$ into an efficiently computable form, we consider so called privacy loss random variables.

**Definition 5.** *Let $P$ and $Q$ be probability density functions. We define the privacy loss function $\mathcal{L}_{P/Q}$ as*

$$\mathcal{L}_{P/Q}(t) = \log \frac{P(t)}{Q(t)}.$$

*We define the privacy loss random variable (PRV) $\omega_{P/Q}$ as*

$$\omega_{P/Q} = \mathcal{L}_{P/Q}(t), \quad t \sim P(t).$$

With slight abuse of notation, we denote the probability density function of the random variable $\omega_{P/Q}$ by $\omega_{P/Q}(t)$, and call it the privacy loss distribution (PLD).

**Theorem 6** (Gopi et al. 15). *The $\delta(\varepsilon)$-bounds can be represented using the following representation that involves the PRV:*

$$H_{\mathrm{e}^\varepsilon}(P \| Q) = \mathop{\mathbb{E}}_{t \sim P} \left[ 1 - \mathrm{e}^{\varepsilon - \mathcal{L}_{P/Q}(t)} \right]_+ = \mathop{\mathbb{E}}_{s \sim \omega_{P/Q}} \left[ 1 - \mathrm{e}^{\varepsilon - s} \right]_+ . \tag{2.1}$$

*Moreover, if $\omega_{P/Q}$ is the PRV for the pair of distributions $(P, Q)$ and $\omega_{P'/Q'}$ the PRV for the pair of distributions $(P', Q')$, then the PRV for the pair of distributions $(P \times P', Q \times Q')$ is given by $\omega_{P/Q} + \omega_{P'/Q'}$.*

When we set $\alpha = \mathrm{e}^\varepsilon$, the following characterisation follows directly from Theorem 6.

**Corollary 7.** *If the pair of distributions $(P, Q)$ is a dominating pair of distributions for a mechanism $\mathcal{M}$, then for all neighbouring datasets $X$ and $X'$ and for all $\varepsilon \in \mathbb{R}$,*

$$\mathop{\mathbb{E}}_{t \sim \mathcal{M}(X)} \left[ 1 - \mathrm{e}^{\varepsilon - \mathcal{L}_{\mathcal{M}(X)/\mathcal{M}(X')}(t)} \right]_+ \leq \mathop{\mathbb{E}}_{t \sim P} \left[ 1 - \mathrm{e}^{\varepsilon - \mathcal{L}_{P/Q}(t)} \right]_+ .$$

We will in particular consider the Gaussian mechanism and its subsampled variant.

**Example: hockey-stick divergence between two Gaussians.** Let $x_0, x_1 \in \mathbb{R}$, $\sigma \geq 0$, and let $P$ be the density function of $\mathcal{N}(x_0, \sigma^2)$ and $Q$ the density function of $\mathcal{N}(x_1, \sigma^2)$. Then, the PRV $\omega_{P/Q}$ is distributed as (Lemma 11 by 30)

$$\omega_{P/Q} \sim \mathcal{N} \left( \frac{(x_0 - x_1)^2}{2\sigma^2}, \frac{(x_0 - x_1)^2}{\sigma^2} \right). \tag{2.2}$$

Thus, in particular: $H_\alpha(P \| Q) = H_\alpha(Q \| P)$ for all $\alpha > 0$. Plugging in PLD $\omega_{P/Q}$ to the expression (2.1), we find that for all $\varepsilon \geq 0$, the hockey-stick $H_{\mathrm{e}^\varepsilon}(P \| Q)$ is given by the expression

$$\delta(\varepsilon) = \Phi \left( -\frac{\varepsilon \sigma}{\Delta} + \frac{\Delta}{2\sigma} \right) - e^\varepsilon \Phi \left( -\frac{\varepsilon \sigma}{\Delta} - \frac{\Delta}{2\sigma} \right), \tag{2.3}$$

where $\Phi$ denotes the CDF of the standard univariate Gaussian distribution and $\Delta = |x_0 - x_1|$. This formula was originally given by Balle and Wang (3).

If $\mathcal{M}$ is of the form $\mathcal{M}(X) = f(X) + Z$, where $f : \mathcal{X}^N \to \mathbb{R}^d$ and $Z \sim \mathcal{N}(0, \sigma^2 I_d)$, and $\Delta = \max_{X \simeq X'} \|f(X) - f(X')\|_2$, then for $x_0 = 0$, $x_1 = \Delta$, $(P, Q)$ of the above form gives a tightly dominating pair of distributions for $\mathcal{M}$ (37). Subsequently, by Theorem 6, $\mathcal{M}$ is $(\varepsilon, \delta)$-DP for $\delta(\varepsilon)$ given by (2.3).

Lemma 8 allows tight analysis of the subsampled Gaussian mechanism using the hockey-stick divergence. We state the result for the case of Poisson subsampling with sampling rate $\gamma$.

**Lemma 8** (Zhu et al. 37)**.** *If $(P, Q)$ dominates a mechanism $\mathcal{M}$ for add neighbors then $(P, (1 - \gamma) \cdot P + \gamma \cdot Q)$ dominates the mechanism $\mathcal{M} \circ S_{Poisson}$ for add neighbors and $((1 - \gamma) \cdot Q + \gamma \cdot P), P)$ dominates $\mathcal{M} \circ S_{Poisson}$ for removal neighbors.*

We will also use the Rényi differential privacy (RDP) (23) which is defined as follows. Rényi divergence of order $\alpha \in (1, \infty)$ between two distributions $P$ and $Q$ is defined as

$$D_\alpha(P\|Q) = \frac{1}{\alpha - 1} \log \int \left( \frac{P(t)}{Q(t)} \right)^\alpha Q(t) \, \mathrm{d}t.$$

By continuity, we have that $\lim_{\alpha \to 1+} D_\alpha(P\|Q)$ equals the KL divergence $KL(P\|Q)$.

**Definition 9.** *We say that a mechanism $\mathcal{M}$ is $(\alpha, \rho)$-RDP, if for all neighbouring datasets $X, X'$, the output distributions $\mathcal{M}(X)$ and $\mathcal{M}(X')$ have Rényi divergence of order $\alpha$ less than $\rho$, i.e.*

$$\max_{X \simeq X'} \{ D_\alpha \big( \mathcal{M}(X) \| \mathcal{M}(X') \big), D_\alpha \big( \mathcal{M}(X') \| \mathcal{M}(X) \big) \} \le \rho.$$

## 2.2 Informal Description: Filtrations, Supermartingales, Stopping Times

Similarly to (34) and (12), we use the notions of filtrations and supermartingales for analysing fully adaptive compositions, where the individual worst-case pairs of distributions are not fixed but can be chosen adaptively based on the outcomes of the previous mechanisms. Given a probability space $(\Omega, \mathcal{F}, \mathbb{P})$, a filtration $(\mathcal{F}_n)_{n \in \mathbb{N}}$ of $\mathcal{F}$ is a sequence of $\sigma$-algebras satisfying: (i) $\mathcal{F}_n \subset \mathcal{F}_{n+1}$ for all $n \in \mathbb{N}$, and (ii) $\mathcal{F}_n \subset \mathcal{F}$ for all $n \in \mathbb{N}$. In the context of the so called natural filtration generated by a stochastic process $X_t$, $t \in \mathbb{N}$, the $\sigma$-algebra of the filtration $\mathcal{F}_n$ represents all the information contained in the outcomes of the first $n$ random variables $(X_1, \ldots, X_n)$. The law of total expectation states that if a random variable $X$ is $\mathcal{F}_n$-measurable and $\mathcal{F}_n \subset \mathcal{F}_{n+1}$, then $\mathbb{E}((X|\mathcal{F}_{n+1})|\mathcal{F}_n) = \mathbb{E}(X|\mathcal{F}_n)$. Thus, if we have natural filtrations $\mathcal{F}_0, \ldots, \mathcal{F}_n$ for a stochastic process $X_0, \ldots, X_n$, then

$$\mathbb{E}(X_n|\mathcal{F}_0) = \mathbb{E}(((X_n|\mathcal{F}_{n-1})|\mathcal{F}_{n-2})|\ldots|\mathcal{F}_0). \tag{2.4}$$

The supermartingale property means that for all $n$,

$$\mathbb{E}(X_n|\mathcal{F}_{n-1}) \le X_{n-1}. \tag{2.5}$$

From the law of total expectation it then follows that for all $n \in \mathbb{N}$, $\mathbb{E}(X_n|\mathcal{F}_0) \le X_0$.

We follow the analysis of Feldman and Zrnic (12) and first set a maximum number of steps (denote by $k$) for the compositions. We do not release more information if a pre-defined privacy budget is exceeded. Informally speaking, the stochastic process $X_n$ that we analyse represents the sum of the realised privacy loss up to step $n$ and the budget remaining at that point. The privacy budget has to be constructed such that the amount of the budget left at step $n$ is included in the filtration $\mathcal{F}_{n-1}$. This allows us to reduce the privacy loss of the adaptively chosen $n$th mechanism from the remaining budget. Mathematically, this means that the integration $\mathbb{E}(X_n|\mathcal{F}_{n-1})$ will be only w.r.t. the outputs of the $n$th mechanism. Consider e.g. the differentially private version of the gradient descend (GD) method, where the amount of increase in the privacy budget depends on the gradient norms which depend on the parameter values at step $n - 1$, i.e., they are included in $\mathcal{F}_{n-1}$. Then, $\mathbb{E}(X_k|\mathcal{F}_0)$ corresponds to the total privacy loss. If we can show that (2.5) holds for $X_n$, then by the law of total expectation the total privacy loss is less than $X_0$, the pre-defined budget. In our case the total budget $X_0$ will equal the $(\varepsilon, \delta)$-curve for a $\mu$-Gaussian DP mechanism, where $\mu$ determines the total privacy budget, and $\mathbb{E}[X_T]$, the expectation of the consumed privacy loss at step $T$, will equal the $(\varepsilon, \delta)$-curve for the fully adaptive composition to be analysed.

A discrete-valued stopping time $\tau$ is a random variable in the probability space $(\Omega, \mathcal{F}, \{\mathcal{F}_t\}, \mathbb{P})$ with values in $\mathbb{N}$ which gives a decision of when to stop. It must be based only on the information present at time $t$, i.e., it has to hold $\{\tau = t\} \in \mathcal{F}_t$. The optimal stopping theorem states that if the stochastic process $X_n$ is a supermartingale and if $T$ is a stopping time, then $\mathbb{E}[X_T] \leq \mathbb{E}[X_0]$. In the analysis of fully adaptive compositions, the stopping time $T$ will equal the step where the privacy budget is about to exceed the limit $B$. Then, only the outputs of the (adaptively selected) mechanisms up to step $T$ are released, and from the optimal stopping theorem it follows that $\mathbb{E}[X_T] \leq X_0$.

# 3 FULLY ADAPTIVE COMPOSITIONS

In order to compute tight $\delta(\varepsilon)$-bounds for fully adaptive compositions, we determine a suitable supermartingale that gives us the analogues of the RDP results of (12).

## 3.1 NOTATION AND THE EXISTING ANALYSIS

Similarly to Feldman and Zrnic (12), we denote the mechanism corresponding to the fully adaptive composition of first $n$ mechanisms as

$$\mathcal{M}^{(n)}(X) = \big(\mathcal{M}_1(X), \mathcal{M}_2(\mathcal{M}_1(X), X), \ldots, \mathcal{M}_n(\mathcal{M}_1(X), \ldots, \mathcal{M}_{n-1}(X), X)\big)$$

and the outcomes of $\mathcal{M}^{(n)}(X)$ as $a^{(n)} = (a_1, \ldots, a_n)$, For datasets $X$ and $X'$, define $\mathcal{L}_{X/X'}^{(n)}$ as

$$\mathcal{L}_{X/X'}^{(n)} = \log\left(\frac{\mathbb{P}(\mathcal{M}^{(n)}(X) = a^{(n)})}{\mathbb{P}(\mathcal{M}^{(n)}(X') = a^{(n)})}\right)$$

and, given $a^{(n-1)}$, we define $\mathcal{L}_{X/X'}^{n}$ as the privacy loss of the mechanism $\mathcal{M}_n$,

$$\mathcal{L}_{X/X'}^{n} = \log\left(\frac{\mathbb{P}(\mathcal{M}_n(a^{(n-1)}, X) = a_n)}{\mathbb{P}(\mathcal{M}_n(a^{(n-1)}, X') = a_n)}\right).$$

Using the Bayes rule it follows that $\mathcal{L}_{X/X'}^{(n)} = \mathcal{L}_{X/X'}^{(n-1)} + \mathcal{L}_{X/X'}^{n} = \sum_{m=1}^{n} \mathcal{L}_{X/X'}^{m}$.

Whitehouse et al. (34) obtain the advanced-composition-like $(\varepsilon, \delta)$-privacy bounds for fully adaptive compositions via a certain privacy loss martingale. However, our approach is motivated by the analysis of Feldman and Zrnic (12). We review the main points of the analysis in Appendix A. The approach of Feldman and Zrnic (12) does not work directly in our case since the hockey-stick divergence does not factorise as the Rényi divergence does. However, we can determine a certain random variable via the hockey-stick divergence and show that it has the desired properties in case the individual mechanisms $\mathcal{M}_i$ have dominating pairs of distributions that are Gaussians. As we show, this requirement is equivalent to them being Gaussian differentially private.

## 3.2 GAUSSIAN DIFFERENTIAL PRIVACY

Informally speaking, a randomised mechanism $\mathcal{M}$ is $\mu$-GDP, $\mu \geq 0$, if for all neighbouring datasets the outcomes of $\mathcal{M}$ are not more distinguishable than two unit-variance Gaussians $\mu$ apart from each other (7). Commonly the Gaussian differential privacy (GDP) is defined using so called trade-off functions (7). For the purpose of this work, we equivalently formalise GDP using pairs of dominating distributions:

**Lemma 10.** *A mechanism $\mathcal{M}$ is $\mu$-GDP, if and only if for all neighbouring datasets $X, X'$ and for all $\alpha > 0$:*

$$H_\alpha(\mathcal{M}(X)||\mathcal{M}(X')) \leq H_\alpha\big(\mathcal{N}(0,1)||\mathcal{N}(\mu,1)\big). \tag{3.1}$$

*Proof.* By Corollary 2.13 of (7), a mechanism is $\mu$-GDP if and only it is $(\varepsilon, \delta)$-DP for all $\varepsilon \geq 0$, where $\delta(\varepsilon) = \Phi\left(-\frac{\varepsilon}{\mu} + \frac{\mu}{2}\right) - e^\varepsilon \Phi\left(-\frac{\varepsilon}{\mu} - \frac{\mu}{2}\right)$. From (2.3) we see that this is equivalent to the fact that for all neighbouring datasets $X, X'$ and for all $\varepsilon \geq 0$: $H_{e^\varepsilon}(\mathcal{M}(X)||\mathcal{M}(X')) \leq H_{e^\varepsilon}(\mathcal{N}(0,1)||\mathcal{N}(\mu,1))$. By Lemma 31 of (37), $H_\alpha\big(\mathcal{M}(X)||\mathcal{M}(X')\big) \leq H_\alpha\big(P, Q\big)$ for all $\alpha > 1$ if and only if $H_\alpha\big(\mathcal{M}(X)||\mathcal{M}(X')\big) \leq H_\alpha\big(Q, P\big)$ for all $0 < \alpha \leq 1$. As $P$ and $Q$ are Gaussians, we see from the form of the privacy loss distribution (2.2) that $H_\alpha\big(Q, P\big) = H_\alpha\big(P, Q\big)$ and that (3.1) holds for all $\alpha > 0$. $\qquad\square$

### 3.3 GDP Analysis of Fully Adaptive Compositions

Analogously to individual RDP parameters (A.1), we define the conditional GDP parameters as

$$\mu_m = \inf\{\mu \geq 0 \, : \, \mathcal{M}^m(\cdot, a^{(m-1)}) \text{ is } \mu-\text{GDP}\}. \tag{3.2}$$

By Lemma 10 above, in particular, this means that for all neighbouring datasets $X, X'$ and for all $\alpha > 0$:

$$H_\alpha(\mathcal{M}^m(X, a^{(m-1)}), \mathcal{M}^m(X', a^{(m-1)}) \leq H_\alpha(\mathcal{N}(\mu_m, 1)||\mathcal{N}(0,1)).$$

Notice that for all $m$, the GDP parameter $\mu_m$ depends on the history $a^{(m-1)}$ and is therefore a random variable, similarly to the conditional RDP values $\rho_m$ defined in (A.1).

**Example: Private GD.** Suppose each mechanism $\mathcal{M}_i$, $i \in [k]$, is of the form $\mathcal{M}_i(X, a) = \sum_{x \in X} f(x, a) + \mathcal{N}(0, \sigma^2)$. Since the hockey-stick divergence is scaling invariant, and since the sensitivity of the deterministic part of $\mathcal{M}_i(X, a)$ is $\max_{x \in X} \|f(x, a^{(m-1)})\|_2$, we have that $\mu_m = \max_{x \in X} \|f(x, a^{(m-1)})\|_2/\sigma$.

We now give the main theorem, which is a GDP equivalent of (Thm. 3.1, 12).

**Theorem 11.** *Let $k$ denote the maximum number of compositions. Suppose that, almost surely,*

$$\sum_{m=1}^{k} \mu_m^2 \leq B^2.$$

*Then, $\mathcal{M}^{(k)}(X)$ is $B$-GDP.*

*Proof.* We here describe the main points, a proof with more details is given in Appendix B. We remark that an alternative proof of this result is given in an independent and concurrent work by Smith and Thakurta (29). First, recall the notation from Section 3.1: $\mathcal{L}_{X/X'}^{(k)}$ denotes the privacy loss between $\mathcal{M}^{(k)}(X)$ and $\mathcal{M}^{(k)}(X')$ with outputs $a^{(k)}$. Let $\varepsilon \in \mathbb{R}$. Our proof is based on showing the supermartingale property for the random variable $M_n(\varepsilon)$, $n \in [k]$, defined as

$$M_k(\varepsilon) = \left[1 - e^{\varepsilon - \mathcal{L}_{X/X'}^{(k)}}\right]_+,$$
$$M_n(\varepsilon) = \mathop{\mathbb{E}}_{t \sim R_n} \left[1 - e^{\varepsilon - \mathcal{L}_{X/X'}^{(n)} - \mathcal{L}_n(t)}\right]_+, \quad 0 \leq n \leq k-1, \tag{3.3}$$

where $\mathcal{L}_n(t) = \log(R_n(t)/Q(t))$ and $R_n$ is the density function of $\mathcal{N}\left(\sqrt{B^2 - \sum_{m=1}^{n} \mu_m^2}, 1\right)$ and $Q$ is the density function of $\mathcal{N}(0, 1)$. This implies that $M_0(\varepsilon) = \mathbb{E}_{t \sim R_0} \left[1 - e^{\varepsilon - \mathcal{L}_0(t)}\right]_+$, where $\mathcal{L}_0(t) = \log(R_0(t)/Q(t))$ and $R_0$ is the density function of $\mathcal{N}(B, 1)$ and $Q$ is the density function of $\mathcal{N}(0, 1)$. In particular, this means that $M_0(\varepsilon)$ gives $\delta(\varepsilon)$ for a $B$-GDP mechanism.

Let $\mathcal{F}_n$ denote the natural filtration $\sigma(a^{(n)})$. First, we need to show that $\mathbb{E}[M_k(\varepsilon)|\mathcal{F}_{k-1}] \leq M_{k-1}(\varepsilon)$. Since the pair of distributions $(\mathcal{N}(\mu_k, 1), \mathcal{N}(0, 1))$ dominates the mechanism $\mathcal{M}_k$, we have by the Bayes rule and Corollary 7,

$$\mathbb{E}\left[M_k(\varepsilon)\Big|\mathcal{F}_{k-1}\right] = \mathop{\mathbb{E}}_{a_k \sim \mathcal{M}_k}\left[\left[1 - e^{\varepsilon - \mathcal{L}_{X/X'}^{(k-1)} - \mathcal{L}_{X/X'}^{k}}\right]_+ \Big|\mathcal{F}_{k-1}\right]$$
$$\leq \mathop{\mathbb{E}}_{t \sim P_k}\left[1 - e^{\varepsilon - \mathcal{L}_{X/X'}^{(k-1)} - \widetilde{\mathcal{L}}_k(t)}\right]_+ \leq M_{k-1}(\varepsilon), \tag{3.4}$$

where $\widetilde{\mathcal{L}}_k(t) = \log(P_k(t)/Q(t))$, $P_k$ is the density function of $\mathcal{N}(\mu_k, 1)$ and $Q$ is the density function of $\mathcal{N}(0, 1)$. Above we have also used the fact that $\mathcal{L}_{X/X'}^{(k-1)} \in \mathcal{F}_{k-1}$. The last inequality follows from the fact that $\sum_{m=1}^{k} \mu_m^2 \leq B^2$ a.s., i.e., $\mu_k \leq (B^2 - \sum_{m=1}^{k-1} \mu_m^2)^{\frac{1}{2}}$ a.s., and from the data-processing inequality. Moreover, we see that $(B^2 - \sum_{m=1}^{k-1} \mu_m^2)^{\frac{1}{2}} \in \mathcal{F}_{k-2}$. Thus we can repeat (3.4) and use the fact that a composition of $\widehat{\mu_1}$-GDP and $\widehat{\mu_2}$-GDP mechanisms is $(\widehat{\mu_1}^2 + \widehat{\mu_2}^2)^{\frac{1}{2}}$-GDP (Cor. 3.3, 7), and by induction see that $M_n(\varepsilon)$ is a supermartingale. By the law of total expectation (2.4), $\mathbb{E}[M_k(\varepsilon)] \leq M_0(\varepsilon)$. By Theorem 6, $\mathbb{E}[M_k(\varepsilon)] = H_{e^\varepsilon}(\mathcal{M}^{(k)}(X)||\mathcal{M}^{(k)}(X'))$, and $M_0(\varepsilon) = H_{e^\varepsilon}(\mathcal{N}(B, 1)||\mathcal{N}(0, 1))$. As $\varepsilon$ was taken to be an arbitrary real number, we see that the inequality $\mathbb{E}[M_k(\varepsilon)] \leq M_0(\varepsilon)$ holds for all $\varepsilon \in \mathbb{R}$ and by Lemma 10, $\mathcal{M}^{(k)}(X)$ is $B$-GDP. $\qquad\square$

## 4 INDIVIDUAL GDP FILTER

Similarly to (12), we can determine an individual GDP privacy filter that keeps track of individual privacy losses and adaptively drops the data elements for which the cumulative privacy loss is about to cross the pre-determined budget (Alg. 1). First, we need to define a GDP filter:

$$\mathcal{F}_B(\mu_1, \dots, \mu_t) = \begin{cases} \text{HALT}, & \text{if } \sum_{i=1}^{t} \mu_i^2 > B^2, \\ \text{CONT}, & \text{else.} \end{cases} \tag{4.1}$$

Also, similarly to (12), we define $\mathcal{S}(x_i, n)$ as the set of dataset pairs $(S, S')$, where $|S| = n$ and $S'$ is obtained from $S$ by deleting the data element $x_i$ from $S$.

---

**Algorithm 1** Individual GDP Filter Algorithm

---

Input: Budget $B$, maximum number of compositions $k$, initial value $a_0$.
**for** $j = 1, \dots, k$ **do**
  For each $i \in [N]$, find $\mu_j^{(i)} \geq 0$ such that for all $\alpha > 0$ and for all $(S, S') \in \mathcal{S}(x_i, n)$:

$$H_\alpha\big(\mathcal{M}_j(S, a^{(j-1)}) || \mathcal{M}_j(S', a^{(j-1)})\big) \leq H_\alpha\big(\mathcal{N}(\mu_j^{(i)}, 1) || \mathcal{N}(0, 1)\big). \tag{4.2}$$

  Define the active set $S_j$: $S_j = \{x_i \ : \ \mathcal{F}_B(\mu_1^{(i)}, \dots, \mu_j^{(i)}) = \text{CONT}\}$.
  For all $x_i$: set $\mu_j^{(i)} = \mu_j^{(i)} \mathbf{1}\{x_i \in S_j\}$.
  Compute $a_j = \mathcal{M}_j(a^{(j-1)}, S_j)$.
**end for**
**return** $a^{(j)}$.

---

Using Theorem 11 and the supermartingale property of a personalised version of $M_n(\varepsilon)$ (Eq. (3.3)), we can show that the output of Alg. 1 is $B$-GDP.

**Theorem 12.** *Denote by $\mathcal{M}$ the output of Algorithm 1. $\mathcal{M}$ is $B$-GDP under remove neighbourhood relation, meaning that for all datasets $X \in \mathcal{X}^N$, for all $i \in [N]$ and for all $\alpha > 0$:*

$$\max\{H_\alpha\big(\mathcal{M}(X) || \mathcal{M}(X^{-i}))\big), H_\alpha\big(\mathcal{M}(X^{-i}) || \mathcal{M}(X))\big)\} \leq H_\alpha\big(\mathcal{N}(B, 1) || \mathcal{N}(0, 1)\big).$$

*Proof.* The proof goes the same way as the proof for (Thm. 4.3 12) which holds for the RDP filter. Let $\mathcal{F}_t$ denote the natural filter $\sigma(a^{(t)})$, and let the privacy filter $\mathcal{F}_B$ be defined as in (4.1). We see that the random variable $T = \min\{\min\{t \ : \ \mathcal{F}_B(\mu_1^{(i)}, \dots, \mu_{t+1}^{(i)}) = \text{HALT}\}, k\}$ is a stopping time since $\{T = t\} \in \mathcal{F}_t$. Let $M_n(\varepsilon)$ be the random variable of Eq. (3.3) defined for the pair of datasets $(X, X^{-i})$ or $(X^{-i}, X)$. From the optimal stopping theorem (26) and the supermartingale property of $M_n(\varepsilon)$ it follows that $\mathbb{E}[M_T(\varepsilon)] \leq M_0(\varepsilon)$ for all $\varepsilon \in \mathbb{R}$. By the reasoning of the proof of Thm.11 we have that Alg. 1 is $B$-GDP. □

### 4.1 BENEFITS OF GDP VS. RDP: MORE ITERATIONS FOR THE SAME PRIVACY

When we replace the RDP filter with a GDP filter for the private GD, we get considerably smaller $\varepsilon$-values. As an example, consider the private GD experiment by Feldman and Zrnic (12) and set $\sigma = 100$ and the number of compositions $k = 420$ (this corresponds to worst-case analysis $\varepsilon = 0.8$ for $\delta = 10^{-5}$). When using GDP instead of RDP, we can run $k = 495$ iterations for an equal value of $\varepsilon$. Figure 3 (Section C.4) depicts the differences in $(\varepsilon, \delta)$-values computed via RDP and GDP.

## 5 APPROXIMATIVE $(\varepsilon, \delta)$-FILTER VIA BLACKWELL'S THEOREM

We next consider a filter that can use any individual dominating pairs of distributions, not just Gaussians. To this end, we need to determine pairs of dominating distributions at each iteration.

**Assumption.** Given neighbouring datasets $X, X'$, we assume that for all $i, i \in [n]$, we can determine a dominating pair of distributions $(P_i, Q_i)$ such that for all $\alpha > 0$,

$$H_\alpha\big(\mathcal{M}_i(a^{(i-1)}, X) || \mathcal{M}_i(a^{(i-1)}, X')\big) \leq H_\alpha\big(P_i || Q_i\big).$$

A tightly dominating pair of distributions $(P_i, Q_i)$ always exists (Proposition 8, 37), and on the other hand, uniquely determining such a pair is straightforward for the subsampled Gaussian mechanism, for example (see Lemma 8). For the so called shufflers, such worst case pairs can be obtained by post-processing (11). As we show in Appendix C.6, the orderings determined by the trade-off functions and the hockey-stick divergence are equivalent. Therefore, from the Blackwell theorem (7, Thm. 2.10) it follows that there exists a stochastic transformation (Markov kernel) $T$ such that $TP_i = \mathcal{M}_i(a^{(i-1)}, X)$ and $TQ_i = \mathcal{M}_i(a^{(i-1)}, X')$.

First, we replace the GDP filter condition $\sum \mu_i^2 \leq B^2$ by the condition $(\mu > 0)$

$$\underset{t_1 \sim P_1, \ldots, t_n \sim P_n}{\mathbb{E}} \left[ 1 - \mathrm{e}^{\varepsilon - \sum_{m=1}^n \mathcal{L}_m(t_m)} \right]_+ \leq H_{\mathrm{e}^\varepsilon} \left( \mathcal{N}(\mu, 1) || \mathcal{N}(0, 1) \right) \tag{5.1}$$

for all $\varepsilon \in \mathbb{R}$. By the Blackwell theorem there exists a stochastic transformation that maps $\mathcal{N}(\mu, 1)$ and $\mathcal{N}(\mu, 0)$ to the product distributions $(P_1, \ldots, P_n)$ and $(Q_1, \ldots, Q_n)$, respectively. From the data-processing inequality for Rényi divergence we then have

$$D_\alpha(P_1 \times \ldots \times P_n || Q_1 \times \ldots \times Q_n) \leq D_\alpha \left( \mathcal{N}(0, 1) || \mathcal{N}(\mu, 1) \right), \tag{5.2}$$

for all $\alpha \geq 1$, where $D_\alpha$ denotes the Rényi divergence of order $\alpha$. Since the pairs $(P_i, Q_i)$, $1 \leq i \leq n$, are the worst-case pairs also for RDP (as described above, due to the data-processing inequality), by (5.2) and the RDP filter results of Feldman and Zrnic (12), we have that for all $\alpha \geq 1$,

$$D_\alpha \left( \mathcal{M}(X) || \mathcal{M}(X') \right) \leq D_\alpha \left( \mathcal{N}(\mu, 1) || \mathcal{N}(0, 1) \right). \tag{5.3}$$

By converting the RDPs of Gaussians in (5.3) to $(\varepsilon, \delta)$-bounds, this procedure provides $(\varepsilon, \delta)$-upper bounds and can be straightforwardly modified into an individual PLD filter as in case of GDP. One difficulty, however, is how to compute the parameter $\mu$ in (5.1), given the individual pairs $(P_i, Q_i)$, $1 \leq i \leq n$. When the number of iterations is large, by the central limit theorem the PLD of the composition starts to resemble that of a Gaussian mechanism (30), and it is then easy to numerically approximate $\mu$ (see Fig. 1 for an example). It is well known that the $(\varepsilon, \delta)$-bounds obtained via RDPs are always non-tight, since the conversion of RDP to $(\varepsilon, \delta)$ is lossy (37). Moreover, often the computation of the RDP values themselves is lossy. In the procedure described here, the only loss comes from converting (5.3) to $(\varepsilon, \delta)$-bounds. In Appendix D we show how to numerically efficiently compute the individual PLDs using FFT.

To illustrate the differences between the individual $\varepsilon$-values obtained with an RDP accountant and with our approximative PLD-based accountant, we consider DP-SGD training of a small feedforward network for MNIST classification. We choose randomly a subset of 1000 data elements and compute their individual $\varepsilon$-values (see Fig. 1). To compute the $\varepsilon$-values, we compare RDP accounting and our approach based on PLDs. We train for 50 epochs with batch size 300, noise parameter $\sigma = 2.0$ and clipping constant $C = 5.0$.

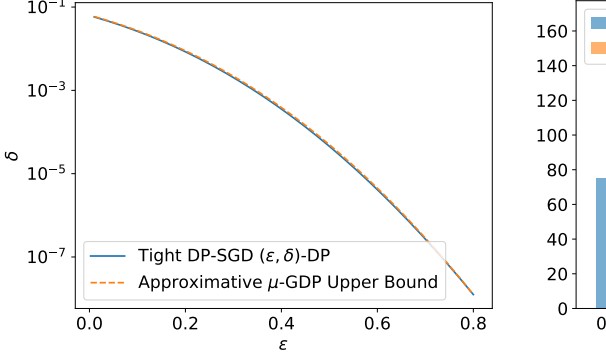 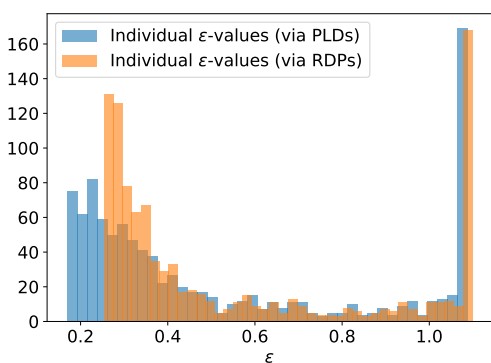

Figure 1: MNIST experiment. Left: Randomly chose data element and its accurate $(\varepsilon, \delta)$-curve after 50 epochs vs. the $\mu$-GDP upper bound approximation. Right: Comparison of individual $\varepsilon$-values obtained via RDPs and PLDs: histograms for randomly selected 1000 samples after 50 epochs ($\delta = 10^{-6}$). Computation using PLDs is better able to capture small individual $\varepsilon$-values.

## 6 EXPERIMENTS WITH MIMIC-III: GROUP-WISE $\varepsilon$-VALUES

For further illustration, we consider the phenotype classification task from a MIMIC-III benchmark library (16) on the clinical database MIMIC-III (17), freely-available from PhysioNet (14). The task is a multi-label classification and aims to predict which of 25 acute care conditions are present in a patient's MIMIC-III record. We have trained a multi-layer perceptron to maximise the macro-averaged AUC-ROC, the task's primary metric. We train the model using DP-GD combined with the Adam optimizer, and use the individual GDP filtering algorithm 1. See Appendix E for further details.

To study the model behaviour between subgroups, we observe five non-overlapping groups of size 1000 from the train set and of size 400 from the test set by the present acute care condition: subgroup 0: no condition at all, subgroups 1 and 2: diagnosed with/not with Pneumonia, subgroups 3 and 4: diagnosed with/not with acute myocardial infarction (heart attack). Similarly as Yu et al. (36), we see a correlation between individual $\varepsilon$-values and model accuracies across the subgroups: the groups with the best privacy protection (smallest average $\varepsilon$-values) have also the smallest average training and test losses. Fig. 2 shows that after running the filtered DP-GD beyond the worst-case $\varepsilon$ - threshold for a number of iterations, both the training and test loss get smaller for the best performing group and larger for other groups. Similarly as DP-SGD has a disparate impact on model accuracies across subgroups (2), we find that while the individual filtering leads to more equal group-wise $\varepsilon$-values, it leads to even larger differences in model accuracies. Here, one could alternatively consider other than algorithmic solutions for balancing the privacy protection among subgroups, by, e.g., collecting more data from the groups with the weakest privacy protection according to the individual $\varepsilon$-values (36). Finally, we observe negligible improvements of the macro-averaged AUC-ROC in the optimal hyperparameter regime using filtered DP-GD, but similarly to (12) improvements can be seen when choosing sub-optimal hyperparameters (see Appendix E.1).

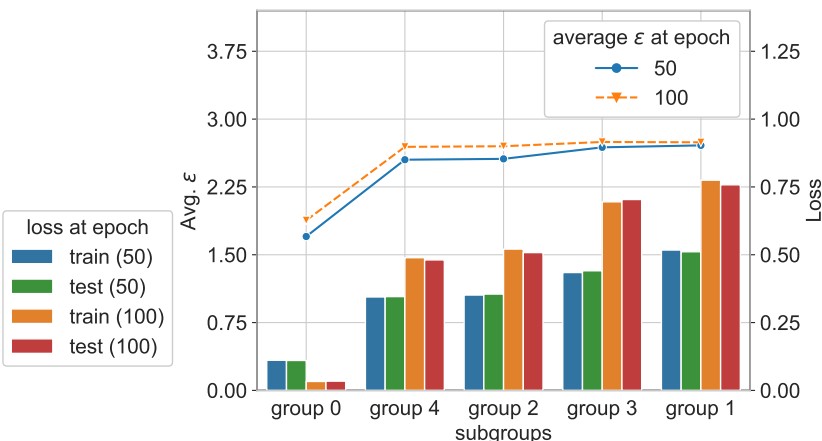

Figure 2: MIMIC III experiment and individual filtering for private GD. Comparing the test losses, training losses and average privacy losses before and after filtering has started (at 50 epochs). The filtering has a further disparate impact on model accuracies across subgroups.

## 7 CONCLUSIONS

To conclude, we have shown how to rigorously carry out fully adaptive analysis and individual DP accounting using the Gaussian DP. We have also proposed an approximative $(\varepsilon, \delta)$-accountant that can utilise any dominating pairs of distributions and shown how to implement it efficiently. As an application we have studied the connection between group-wise individual privacy parameters and model accuracies when using DP-GD, and found that the filtering further amplifies the model accuracy imbalance between groups. An open question remains how to carry out tight fully adaptive analysis using arbitrary dominating pairs of distributions.

## 8 ETHICS STATEMENT

Our work is on improving differential privacy techniques, which contributes to the strong theoretical foundation of privacy-preserving machine learning, an essential component of trustworthy machine learning. Our method provides accurate estimates of individual privacy loss, therefore helping to evaluate the impact of privacy-preserving machine learning to individual privacy. Our experiments indicate that the filtered DP gradient descent has disparate impact on subgroups of data, and should therefore be used with caution.

Our experiments use the MIMIC-III data set of pseudonymised health data by permission of the data providers. The data was processed according to usage rules defined by the data providers, and all reported results are anonymised. All the code related to MIMIC-III data set is publicly available (`https://github.com/DPBayes/individual-accounting-gdp`), as requested by Physionet (`https://physionet.org/content/mimiciii/view-dua/1.4/`).

## 9 REPRODUCIBILITY STATEMENT

All the missing details from proofs and missing proofs in the main text are given in the Appendix, as well as more detailed descriptions of the experiments. Pythons codes needed for the experiments and plots is made publicly available (`https://github.com/DPBayes/individual-accounting-gdp`). To compute $(\varepsilon, \delta)$-bounds via RDP, we have used the conversion formula of Canonne et al. (4). GDP bounds are easily converted to $(\varepsilon, \delta)$-bounds using the formula (C.2).

## 10 ACKNOWLEDGMENTS

The authors acknowledge CSC – IT Center for Science, Finland, and the Finnish Computing Competence Infrastructure (FCCI) for computational and data storage resources.

This work was supported by the Academy of Finland (Flagship programme: Finnish Center for Artificial Intelligence, FCAI; and grant 325573), the Strategic Research Council at the Academy of Finland (Grant 336032) as well as the European Union (Project 101070617). Views and opinions expressed are however those of the author(s) only and do not necessarily reflect those of the European Union or the European Commission. Neither the European Union nor the granting authority can be held responsible for them.

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

## A  EXISTING ANALYSIS USING RDP BY FELDMAN AND ZRNIC (12)

We next illustrate how the stochastic process $X_n$ that is used to analyse fully adaptive compositions is determined in case of RDP analysis (12). Central in the analysis is showing the supermartingale property (2.5) of $X_n$.

The fully adaptive RDP analysis by Feldman and Zrnic (12) is based on studying the properties of the supermartingale $M_n$ which they define as

$$M_n(X, X') = \text{Loss}(a^{(n)}, X, X', \alpha) \cdot e^{-(\alpha-1)\sum_{m=1}^n \rho_m},$$

where $\alpha \geq 1$,

$$\text{Loss}(a^{(n)}, X, X', \alpha) = \left( \frac{\mathbb{P}(\mathcal{M}^{(n)}(X) = a^{(n)})}{\mathbb{P}(\mathcal{M}^{(n)}(X') = a^{(n)})} \right)^\alpha,$$

and $\rho_m$ gives the RDP of order $\alpha$ given $\mathcal{M}_{1:m-1}(X)$, i.e.

$$\rho_m = \frac{1}{\alpha - 1} \log \sup_{(X,X') \in \mathcal{S}} \mathbb{E}_{a^{(m)} \sim \mathcal{M}^{(m)}(X')} \left[ \left( \frac{\mathbb{P}(\mathcal{M}^{(m)}(X) = a^{(m)})}{\mathbb{P}(\mathcal{M}^{(m)}(X') = a^{(m)})} \right)^{\alpha} \Bigg| a^{(m-1)} \right], \quad \text{(A.1)}$$

where $\mathcal{S}$ is a pre-determined set of neighbouring datasets In particular, the RDP bounds for the fully adaptive compositions are obtained by showing that $M_n(X, X')$ has the supermartingale property, meaning that

$$\mathbb{E}(M_n(X, X')|\mathcal{F}_{n-1}) \leq M_{n-1}(X, X'). \quad \text{(A.2)}$$

Feldman and Zrnic (12) show that from this property, and from the law of total expectation (2.4), it follows that if $\sum_{i=1}^{k} \rho_i \leq B$ almost surely, where $K$ is the maximum number of compositions, then the fully adaptive composition is $(\alpha, B)$-RDP (Thm. 3.1, 12) .

Due to the factorisability of the Rényi divergence, the property (A.2) is straightforward to show for the random variable $M_n(X, X')$ using the Bayes theorem:

$$
\begin{aligned}
&\mathbb{E}(M_n(X, X')|\mathcal{F}_{n-1}) \\
&= \mathbb{E}(\text{Loss}(a^{(n)}, X, X', \alpha) \cdot e^{-(\alpha-1)\sum_{m=1}^{n} \rho_m} | \mathcal{F}_{n-1}) \\
&= \mathbb{E} \left( \frac{\mathbb{P}(\mathcal{M}_n(X) = a_n | a^{(n-1)})}{\mathbb{P}(\mathcal{M}_n(X') = a_n | a^{(n-1)})} \right)^{\alpha} \cdot \text{Loss}(a^{(n-1)}, X, X', \alpha) \cdot e^{-(\alpha-1)\sum_{m=1}^{n} \rho_m},
\end{aligned}
\quad \text{(A.3)}
$$

since $\rho_1, \ldots, \rho_n \in \mathcal{F}_{n-1}$ and $\text{Loss}(a^{(n-1)}, X, X', \alpha) \in \mathcal{F}_{n-1}$. Moreover, as $\mathcal{M}_n$ is $\rho_n$-RDP,

$$\text{Loss}(a^{(n-1)}, X, X', \alpha) \leq e^{(\alpha-1)\rho_n},$$

and the supermartingale property follows from (A.3), i.e., that

$$\mathbb{E}(M_n(X, X')|\mathcal{F}_{n-1}) \leq M_{n-1}(X, X').$$

As the hockey-stick divergence does not factorise in this way, we need to take another approach to get the required supermartingale.

## B  MAIN THEOREM

**Theorem B.1.** *Let $k$ denote the maximum number of compositions. Suppose that, almost surely,*

$$\sum_{m=1}^{k} \mu_m^2 \leq B^2. \quad \text{(B.1)}$$

*Then, $\mathcal{M}^{(k)}(X)$ is $B$-GDP.*

*Proof.* First, recall the notation from Section 3.1: $\mathcal{L}_{X/X'}^{(k)}$ denotes the privacy loss between $\mathcal{M}^{(k)}(X)$ and $\mathcal{M}^{(k)}(X')$ with outputs $a^{(k)}$. Let $\varepsilon \in \mathbb{R}$. Our proof is based on showing the supermartingale property for the random variable $M_n(\varepsilon)$, $n \in [k]$, defined as

$$
\begin{aligned}
M_k(\varepsilon) &= \left[ 1 - e^{\varepsilon - \mathcal{L}_{X/X'}^{(k)}} \right]_+, \\
M_n(\varepsilon) &= \mathbb{E}_{t \sim R_n} \left[ 1 - e^{\varepsilon - \mathcal{L}_{X/X'}^{(n)} - \mathcal{L}_n(t)} \right]_+, \quad 0 \leq n \leq k - 1,
\end{aligned}
\quad \text{(B.2)}
$$

where $\mathcal{L}_n(t) = \log R_n(t)/Q(t)$ and $R_n$ is the density function of $\mathcal{N}\left(\sqrt{B^2 - \sum_{m=1}^{n} \mu_m^2}, 1\right)$ and $Q$ is the density function of $\mathcal{N}(0, 1)$. Moreover,

$$M_0(\varepsilon) = \mathbb{E}_{t \sim R_0} \left[ 1 - e^{\varepsilon - \mathcal{L}_0(t)} \right]_+,$$

where $\mathcal{L}_0(t) = \log R_0(t)/Q(t)$ and $R_0$ is the density function of $\mathcal{N}(B, 1)$ and $Q$ is the density function of $\mathcal{N}(0, 1)$. Notice that in particular this means that $M_0(\varepsilon)$ gives $\delta(\varepsilon)$ for a $B$-GDP mechanism.

We next show that $\mathbb{E}\left[M_k(\varepsilon)\Big|\mathcal{F}_{k-1}\right] \leq M_{k-1}(\varepsilon)$ for all $k$. The supermartingale property follows then by induction.

Since the pair of distributions $\left(\mathcal{N}(\mu_k, 1), \mathcal{N}(0,1)\right)$ dominates the mechanism $\mathcal{M}_k$, we have by the Bayes rule and Corollary 7 that

$$
\begin{aligned}
\mathbb{E}\left[M_k(\varepsilon)\Big|\mathcal{F}_{k-1}\right] &= \underset{a^{(k)}\sim\mathcal{M}^{(k)}}{\mathbb{E}}\left[\left[1 - e^{\varepsilon - \mathcal{L}_{X/X'}^{(k)}}\right]_+ \Bigg|\mathcal{F}_{k-1}\right] \\
&= \underset{a^k\sim\mathcal{M}_k(\varepsilon)}{\mathbb{E}}\left[\left[1 - e^{\varepsilon - \mathcal{L}_{X/X'}^{(k-1)} - \mathcal{L}_{X/X'}^{k}}\right]_+ \Bigg|\mathcal{F}_{k-1}\right] \\
&\leq \underset{t\sim P_k}{\mathbb{E}}\left[1 - e^{\varepsilon - \mathcal{L}_{X/X'}^{(k-1)} - \widetilde{\mathcal{L}}_k(t)}\right]_+,
\end{aligned}
$$

where $\widetilde{\mathcal{L}}_k(t) = \log P_k(t)/Q(t)$ and $P_k$ is the density function of $\mathcal{N}(\mu_k, 1)$ and $Q$ is the density function of $\mathcal{N}(0,1)$. Above we have also used the fact that $\mathcal{L}_{X/X'}^{(k-1)} \in \mathcal{F}_{k-1}$.

Since $\sum_{m=1}^{k}\mu_m^2 \leq B^2$ almost surely, i.e., $\mu_k \leq \sqrt{B^2 - \sum_{m=1}^{k-1}\mu_m^2}$ almost surely, by the data-processing inequality for $\alpha$-divergence we have that, almost surely,

$$
\underset{t\sim P_k}{\mathbb{E}}\left[1 - e^{\varepsilon - \mathcal{L}_{X/X'}^{(k-1)} - \widetilde{\mathcal{L}}_k(t)}\right]_+ \leq \underset{t\sim R_{k-1}}{\mathbb{E}}\left[1 - e^{\varepsilon - \mathcal{L}_{X/X'}^{(k-1)} - \mathcal{L}_{k-1}(t)}\right]_+ = M_{k-1}(\varepsilon),
$$

where $\mathcal{L}_{k-1}(t) = \log R_{k-1}(t)/Q(t)$ and $R_{k-1}$ is the density function of $\mathcal{N}\left(\sqrt{B^2 - \sum_{m=1}^{k-1}\mu_m^2}, 1\right)$ and $Q$ is the density function of $\mathcal{N}(0,1)$. Therefore, $\mathbb{E}\left[M_k(\varepsilon)\Big|\mathcal{F}_{k-1}\right] \leq M_{k-1}(\varepsilon)$.

Since $\mu_1, \ldots, \mu_{k-1} \in \mathcal{F}_{k-2}$, we have that, almost surely,

$$
\begin{aligned}
\mathbb{E}\left[M_{k-1}(\varepsilon)\Big|\mathcal{F}_{k-2}\right] &= \underset{a^{(k-1)}\sim\mathcal{M}^{(k-1)}}{\mathbb{E}}\left[\underset{t\sim R_{k-1}}{\mathbb{E}}\left[1 - e^{\varepsilon - \mathcal{L}_{X/X'}^{(k-1)} - \mathcal{L}_{k-1}(t)}\right]_+ \Bigg|\mathcal{F}_{k-2}\right] \\
&= \underset{a^{k-1}\sim\mathcal{M}_{k-1}}{\mathbb{E}}\left[\underset{t\sim R_{k-1}}{\mathbb{E}}\left[1 - e^{\varepsilon - \mathcal{L}_{X/X'}^{(k-2)} - \mathcal{L}_{X/X'}^{k-1} - \mathcal{L}_{k-1}(t)}\right]_+ \Bigg|\mathcal{F}_{k-2}\right] \\
&= \underset{t\sim R_{k-1}}{\mathbb{E}}\left[\underset{a^{k-1}\sim\mathcal{M}_{k-1}}{\mathbb{E}}\left[1 - e^{\varepsilon - \mathcal{L}_{X/X'}^{(k-2)} - \mathcal{L}_{X/X'}^{k-1} - \mathcal{L}_{k-1}(t)}\right]_+ \Bigg|\mathcal{F}_{k-2}\right] \quad\text{(B.3)} \\
&\leq \underset{t\sim R_{k-1}}{\mathbb{E}}\ \underset{t_{k-1}\sim P_{k-1}}{\mathbb{E}}\left[1 - e^{\varepsilon - \mathcal{L}_{X/X'}^{(k-2)} - \widetilde{\mathcal{L}}_{k-1}(t_{k-1}) - \mathcal{L}_{k-1}(t)}\right]_+ \\
&= \underset{t\sim R_{k-2}}{\mathbb{E}}\left[1 - e^{\varepsilon - \mathcal{L}_{X/X'}^{(k-2)} - \mathcal{L}_{k-2}(t)}\right]_+ \\
&= M_{k-2},
\end{aligned}
$$

where $\widetilde{\mathcal{L}}_{k-1}(t) = \log P_{k-1}(t)/Q(t)$ and $P_{k-1}$ is the density function of $\mathcal{N}\left(\mu_{k-1}, 1\right)$ and $Q$ is the density function of $\mathcal{N}(0,1)$. In the inequality step we use Corollary 7 and the fact that the pair of distributions $\left(\mathcal{N}(\mu_{k-1}, 1), \mathcal{N}(0,1)\right)$ dominates the mechanism $\mathcal{M}_{k-1}(\varepsilon)$. In the second last step we have also use the fact that if $\widehat{P}_1 \sim \mathcal{N}(\widehat{\mu}_1, 1)$, $\widehat{P}_1 \sim \mathcal{N}(\widehat{\mu}_2, 1)$ and $Q \sim \mathcal{N}(0,1)$, and $\widehat{\mathcal{L}}_1(t) = \log \widehat{P}_1(t)/Q(t)$ and $\widehat{\mathcal{L}}_2(t) = \log \widehat{P}_2(t)/Q(t)$, then

$$
\underset{t_1\sim\widehat{P}_1}{\mathbb{E}}\ \underset{t_2\sim\widehat{P}_2}{\mathbb{E}}\left[1 - e^{\varepsilon - \widehat{\mathcal{L}}_1(t) - \widehat{\mathcal{L}}_2(t)}\right]_+ = \underset{t\sim\widehat{P}_3}{\mathbb{E}}\left[1 - e^{\varepsilon - \widehat{\mathcal{L}}_3(t)}\right]_+,
$$

where $\widehat{P}_3 \sim \mathcal{N}(\sqrt{\widehat{\mu}_1^2 + \widehat{\mu}_2^2}, 1)$ and $\widehat{\mathcal{L}}_3(t) = \log \widehat{P}_3(t)/Q(t)$. This follows directly from the fact that the PLDs determined by the pairs of distributions $(\widehat{P}_1, Q)$ and $(\widehat{P}_2, Q)$ are Gaussians (see Eq. (2.2)), the convolution of two Gaussians is a Gaussian.

By induction, we see from (B.3) that the the supermartingale property holds for the random variable $M_n(\varepsilon)$. By the law of total expectation (2.4), $\mathbb{E}[M_k(\varepsilon)] \leq M_0(\varepsilon)$. By Theorem 6,

$$\mathbb{E}[M_k(\varepsilon)] = H_{e^\varepsilon}\big(\mathcal{M}^{(k)}(X)||\mathcal{M}^{(k)}(X')\big),$$

and

$$M_0(\varepsilon) = H_{e^\varepsilon}\big(\mathcal{N}(B,1)||\mathcal{N}(0,1)\big).$$

As $\varepsilon$ was taken to be an arbitrary real number, the inequality $\mathbb{E}[M_k(\varepsilon)] \leq M_0(\varepsilon)$ holds for all $\varepsilon \in \mathbb{R}$ and by Lemma 10 we see that $\mathcal{M}^{(k)}(X)$ is $B$-GDP. $\qquad \square$

## C  FILTERS AND ODOMETERS

We here give additional details on the GDP filters and shortly discuss implementation of GDP privacy odometers as well.

### C.1  GDP - PRIVACY FILTER

For simplicity, we here consider a GDP filter that chooses the privacy parameters adaptively, but not individually (like the filter in the main text). I.e., the amount that the privacy budget is spent at each step has to provide a guarantee over the whole dataset.

To this end we formally define a GDP filter as

$$\mathcal{F}_B(\mu_1,\ldots,\mu_t) = \begin{cases} \text{HALT}, & \text{if } \sum_{i=1}^t \mu_i^2 > B^2, \\ \text{CONT}, & \text{else.} \end{cases}$$

Using the filter $\mathcal{F}_B$, a GDP filter is given as in Alg. 2.

---

**Algorithm 2** GDP Filter Algorithm

---

Input: Budget $B$, maximum number of compositions $k$, initial value $a_0$.
**for** $j = 1,\ldots,k$ **do**
    Find parameter $\mu_j \geq 0$ such that $\mathcal{M}_j(\cdot, a^{(j-1)})$ is $\mu_j$-GDP.
    **if** $\sum_{\ell=0}^j \mu_\ell^2 > B^2$: **then**
      BREAK
    **else**
      $a_j = \mathcal{M}_j(X, a^{(j-1)})$
    **end if**
**end for**
**return** $a^{(j-1)}$.

---

In principle, the supermartingale property of the random variable $M_n(\varepsilon)$, as defined in (3.3), is sufficient to show that the algorithm below is $B$-GDP. The only difference is that the algorithm can stop at random time. To include that feature in the analysis, we need to use the optimal stopping time theorem.

**Theorem C.1.** *Denote by $\mathcal{M}$ the output of Algorithm 2. $\mathcal{M}$ is $B$-GDP under remove neighbourhood relation, meaning that for all datasets $X \in \mathcal{X}^N$, for all $i \in [N]$ and for all $\alpha > 0$:*

$$\max\{H_\alpha\big(\mathcal{M}(X)||\mathcal{M}(X^{-i}))\big), H_\alpha\big(\mathcal{M}(X^{-i})||\mathcal{M}(X))\big)\} \leq H_\alpha\big(\mathcal{N}(B,1)||\mathcal{N}(0,1)\big). \quad \text{(C.1)}$$

*Proof.* The proof goes exactly the same as the proof for (Thm. 4.3 12) which holds for the RDP filter. By using the fact that for all $t \geq 0$: $\mu_{t+1} \in \mathcal{F}_t$, where $\mathcal{F}_t$ is the natural filter $\sigma(a^{(t)})$, we see that the random variable

$$T = \min\{t : \mathcal{F}_B(\mu_1,\ldots,\mu_{t+1}) = \text{HALT}\} \wedge k$$

is a stopping time since $\{T = t\} \in \mathcal{F}_t$ since $\mu_{t+1} \in \mathcal{F}_t$. Let $M_n(\varepsilon)$ be the random variable of Eq. (3.3) defined for the pair of datasets $(X, X^{-i})$ or $(X^{-i}, X)$. From the optimal stopping theorem and the supermartingale property it then follows that for all $\varepsilon \in \mathbb{R}$, $\mathbb{E}[M_T(\varepsilon)] \leq M_0(\varepsilon)$, which by the reasoning of the proof of Thm.11 shows that (C.1) holds, i.e., output of Alg. 2 is $B$-GDP w.r.t. to the removal neighbourhood relation of datasets. $\qquad \square$

A benefit of GDP filter when compared to RDP filter is that we obtain tight $(\varepsilon, \delta(\varepsilon))$-bounds for adaptive compositions of Gaussian mechanisms. Moreover, from Thm. 10 it follows that these tight $(\varepsilon, \delta(\varepsilon))$-DP bounds can be obtained by an analytic formula:

**Corollary C.2.** *For GDP-budget $B > 0$, the outputs of Algorithms 1 and 2 are $(\varepsilon, \delta(\varepsilon))$-DP for all $\varepsilon \geq 0$ and*

$$\delta(\varepsilon) = \Phi\left(-\frac{\varepsilon}{B} + \frac{B}{2}\right) - e^\varepsilon \, \Phi\left(-\frac{\varepsilon}{B} - \frac{B}{2}\right). \tag{C.2}$$

## C.2 GDP Parameters for the Individual Filtering of Private GD

Notice that we have the following for the individual GDP filtering. Suppose each mechanism $\mathcal{M}_i$, $i \in [k]$, in the sequence is of the form

$$\mathcal{M}_i(X, a) = \sum_{x \in X} f(x, a) + \mathcal{N}(0, \sigma^2).$$

Since the hockey-stick divergence is scaling invariant and since the sensitivity of $\sum_{x \in X} f(x, a^{(j-1)})$ w.r.t. to removal of $x_i$ is $\|f(x_i, a^{(j-1)})\|_2$, we have that

$$\mu_j^{(i)} = \|f(x_i, a^{(j-1)})\|_2 / \sigma.$$

## C.3 Tight Bounds for the Gaussian Mechanism

When running e.g. the DP-GD algorithm and using either the filtering of Alg. 2 or the individual filtering of Alg. 1, by appropriate scaling of the gradients each individual data element can be made to fully consume its privacy budget. This scaling for individual filtering is given in (Algorithm 3 12).

**Remark C.3.** *Suppose we use the Gaussian mechanism and scale the noise for each data element $x_i$, $i \in [N]$ at the last step such that the GDP budget is fully consumed, i.e., we have that $\sum_j \mu_j^{(i)} = B$, then the resulting algorithm is tightly $(\varepsilon, \delta)$-DP for $\delta(\varepsilon)$ given by the expression (C.2), in a sense that for all $i \in [N]$,*

$$\max\{H_{e^\varepsilon}\big(\mathcal{M}^{(k)}(X) \| \mathcal{M}^{(k)}(X^{-i})\big)\big), H_\alpha\big(\mathcal{M}^{(k)}(X^{-i}) \| \mathcal{M}^{(k)}(X)\big)\big)\} = \delta(\varepsilon).$$

## C.4 Benefits of GDP vs. RDP Filtering

To experimentally illustrate the benefits of GDP accounting, consider one of the private GD experiments of (12), where $\sigma = 100$, and number of compositions corresponding to worst-case analysis is $k = 420$. The RDP value of order $\alpha$ corresponding to this iteration is then $\alpha/(2 \cdot \widetilde{\sigma}^2)$, where $\widetilde{\sigma} = \sigma/\sqrt{k}$. Figure 3 shows the $(\varepsilon, \delta)$-values, computed via RDP and GDP. To get the $(\varepsilon, \delta)$-values from the RDP-values, we use the conversion formula of Lemma C.4 below. When using GDP instead of RDP, we can run $k = 495$ iterations instead of the $k = 420$ iterations, for an equal privacy budget of $\varepsilon = 0.8$, when $\delta = 10^{-5}$.

Rényi DP parameters are converted to $(\varepsilon, \delta)$-DP by minimizing w.r.t. $\lambda$ over the values given by (C.3).

**Lemma C.4** (Canonne et al. 4). *Suppose the mechanism $\mathcal{M}$ is $(\lambda, \varepsilon')$-RDP. Then $\mathcal{M}$ is also $(\varepsilon, \delta(\varepsilon))$-DP for arbitrary $\varepsilon \geq 0$ with*

$$\delta(\varepsilon) = \frac{\exp\big((\lambda - 1)(\varepsilon' - \varepsilon)\big)}{\lambda} \left(1 - \frac{1}{\lambda}\right)^{\lambda - 1}. \tag{C.3}$$

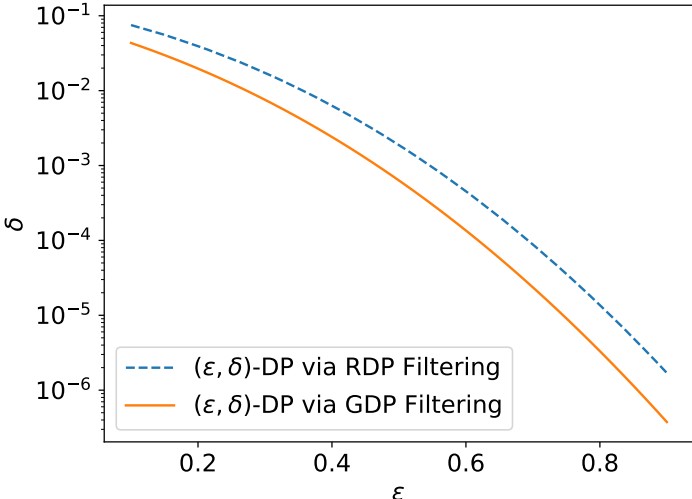

Figure 3: Comparison of RDP and GDP for private GD with $\sigma = 100$ and number of iterations $k = 420$ (experiment conisdered by Feldman and Zrnic (12)). This means that when we replace the RDP filter with GDP filter for the private GD, we can get roughly 10 percent smaller $\varepsilon$-values.

### C.4.1 FURTHER COMPARISONS OF RDP AND GDP

Figures 4 and 5 further illustrate the differences between RDP and GDP accounting for filtering. Figure 4 shows the effect of number of compositions $k$, when $\sigma = 100$ and Figure 5 illustrates the maximum number of compositions for a given privacy budget $\varepsilon$, when $\sigma = 100$ and $\delta = 10^{-5}$.

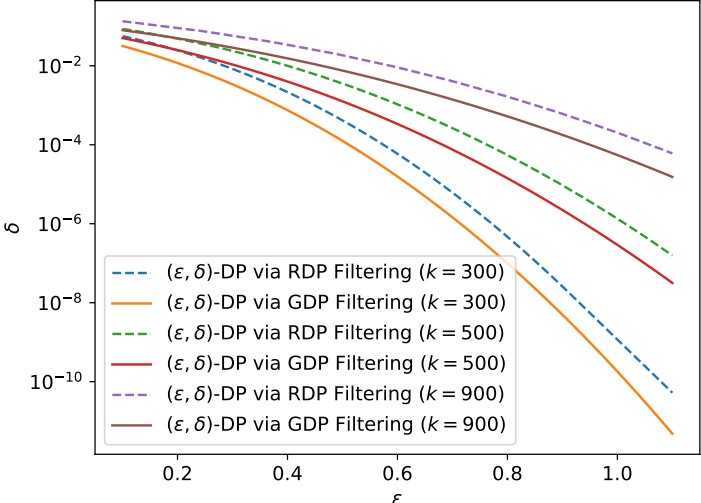

Figure 4: Comparison of RDP and GDP for private GD with $\sigma = 100$ and different number of compositions $k$.

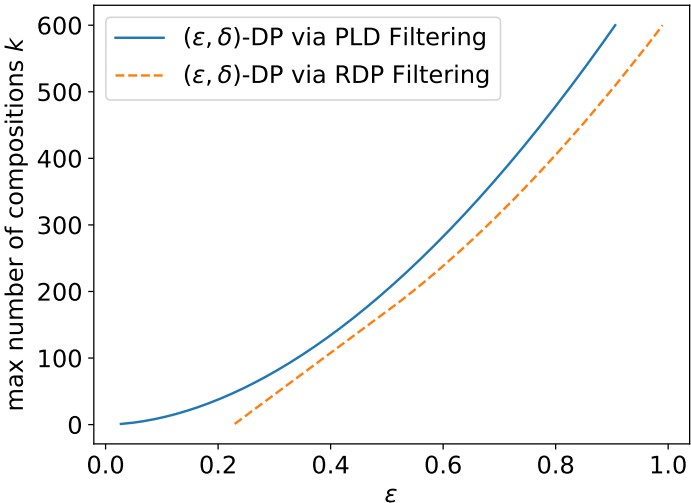

Figure 5: Comparison of RDP and GDP for private GD with $\sigma = 100$: maximum number of allowed steps for private GD (number of compositions $k$) for different values of $\varepsilon$, when $\delta = 10^{-5}$.

### C.5 GDP PRIVACY ODOMETERS

We here also shortly comment on privacy odometers, considered e.g. by (28; 12; 21).

In practice, one might want to track the privacy loss incurred so far. Rogers et al. (28) were the first ones to formalise this in terms of a privacy odometer. Feldman and Zrnic (12) utilise a sequence of valid Rényi privacy filters such that a fixed sequence of privacy losses $B_1, B_2, \ldots$ determine random stopping times $T_1, T_2, \ldots$ such that the privacy spent up to time $T_i$ is at most $B_i$. By assuming, for example, that for all $i$, $B_{i+1} - B_i = \Delta$ for a fixed discretisation parameter $\Delta > 0$, we may employ the RDP filter such that whenever the privacy budget counter crosses $\Delta$ (suppose for the $m$th time) we release the sequence $a^{(T_m)}$ and initialize the privacy loss counter to zero. The fact that $a^{(T_m)}$ is $m\Delta$-RDP follows directly from the RDP results that hold for the filters.

With the GDP, we can construct in the exactly same way an algorithm that outputs states always after every predetermined amount of GDP budget $\Delta$ is spent. If at round $i$ we spend $\Delta_i$-GDP budget, by the results for GDP filters we know that $B_{i+1}^2 = B_i^2 + \Delta_i^2$, and that the output $a^{(T_m)}$ is $\widetilde{B}_m$-GDP, where $\widetilde{B}_m = \sqrt{\Delta_1^2 + \ldots + \Delta_m^2}$.

### C.6 BLACKWELL'S THEOREM VIA DOMINATING PAIRS OF DISTRIBUTIONS

There is a one-to-one relationship between the orderings determined by trade-off functions and the hockey-stick divergence and this follows from the results by Zhu et al. (37) as follows. Let $P$ and $Q$ be probability distributions and denote by $T(P, Q)$ the trade-off function determined by $P$ and $Q$ and let $H_\alpha(P||Q)$ denote the hockey-stick divergence or order $\alpha > 0$. The Lemma 20 of (37) is a restatement of (Proposition 2.12, 7) and it states that for any $\varepsilon \in \mathbb{R}$,

$$H_{e^\varepsilon}(P||Q) = 1 + T[P, Q]^*(-e^\varepsilon), \tag{C.4}$$

where, for a trade-off function $f$, $f^*$ denotes the function

$$f^*(y) = \sup_{x \in \mathbb{R}} yx - f(x).$$

From (C.4) the equivalence follows directly, i.e., if $(\widetilde{P}, \widetilde{Q})$ is another pair of probability distributions, then

$$H_{e^\varepsilon}(\widetilde{P}||\widetilde{Q}) \leq H_{e^\varepsilon}(P||Q)$$

for all $\varepsilon \in \mathbb{R}$ if and only if

$$T[\widetilde{P}, \widetilde{Q}] \geq T[P, Q].$$

This also means that, if

$$H_\alpha(\widetilde{P}||\widetilde{Q}) \leq H_\alpha(P||Q)$$

for all $\alpha > 0$, then by the Blackwell theorem (see e.g. Thm. 2.10, 7), there exists a stochastic transformation (Markov kernel) $T$ such that $TP = \widetilde{P}$ and $TQ = \widetilde{Q}$.

## D  EFFICIENT INDIVIDUAL NUMERICAL ACCOUNTING FOR DP-SGD

We next show how to compute the individual PLDs for DP-SGD. These are needed when implementing the approximative individual $(\varepsilon, \delta)$-accountant described in Section 5. The errors arising from the approximations are generally negligible and a rigorous error analysis could be carried out using the techniques presented in (20) and (15).

The numerical approximation is based on

1. a numerical $\sigma$-grid which allows evaluating upper bounds for $\delta$'s efficiently: we precompute FFTs for different $\sigma$-values and no additional FFT computations are then needed during the evaluation of the individual $\varepsilon$-values. By the data-processing inequality this grid approximation also leads to upper $(\varepsilon, \delta)$-bounds.
2. the Plancherel Theorem, which removes the need to compute inverse FFTs when evaluating individual PLDs.

First, we recall some basics about numerical accounting using FFT (see also (19; 15)).

### D.1  NUMERICAL EVALUATION OF DP PARAMETERS USING FFT

We use a Fast Fourier Transform (FFT)-based method by Koskela et al. (19; 20) called the Fourier Accountant (FA). The same approximation could be done when using the PRV accountant by Gopi et al. (15). Using FFT requires that we truncate and place the PLD $\omega$ on an equidistant numerical grid over an interval $[-L, L]$, $L > 0$. Convolutions are evaluated using the FFT algorithm, and using the existing error analysis (see e.g., 20), the error incurred by the numerical FFT approximation can be bounded.

The Fast Fourier Transform (FFT) is described as follows (5). Let $x = [x_0, \ldots, x_{n-1}]^{\mathrm{T}}$, $w = [w_0, \ldots, w_{n-1}]^{\mathrm{T}} \in \mathbb{R}^n$. The discrete Fourier transform $\mathcal{F}$ and its inverse $\mathcal{F}^{-1}$ are defined as (31)

$$(\mathcal{F}x)_k = \sum_{j=0}^{n-1} x_j \mathrm{e}^{-\mathrm{i}\,2\pi kj/n}, \quad (\mathcal{F}^{-1}w)_k = \frac{1}{n}\sum_{j=0}^{n-1} w_j \mathrm{e}^{\mathrm{i}\,2\pi kj/n},$$

where $\mathrm{i} = \sqrt{-1}$. Using FFT the running time of evaluating $\mathcal{F}x$ and $\mathcal{F}^{-1}w$ reduces to $O(n \log n)$. Also, FFT enables evaluating discrete convolutions efficiently using the so called convolution theorem

For obtaining computational speed-ups, we use the Plancherel Theorem (Chpt. 12, 25), which states that the DFT preserves inner products: for $x, y \in \mathbb{R}^n$,

$$\langle x, y \rangle = \frac{1}{n}\langle \mathcal{F}(x), \mathcal{F}(y)\rangle.$$

When using FA to approximate $\delta(\varepsilon)$, we need to evaluate an expression of the form

$$\boldsymbol{b}^k = D\,\mathcal{F}^{-1}\big(\mathcal{F}(D\boldsymbol{a}^1)^{\odot k_1} \odot \cdots \odot \mathcal{F}(D\boldsymbol{a}^m)^{\odot k_m}\big), \quad D = \begin{bmatrix} 0 & I_{n/2} \\ I_{n/2} & 0 \end{bmatrix} \in \mathbb{R}^{n \times n},$$

where $\boldsymbol{a}^i$ corresponds to a numerical PLD for a combination of DP hyperparameters $i$, and $k_i$ is the number of times the composition contains a mechanism with this PLD.

Approximation for $\delta(\varepsilon)$ is then obtained from the discrete sum that approximates the hockey-stick integral:

$$\widetilde{\delta}(\varepsilon) = \sum_{-L+\ell\Delta x > \varepsilon} \big(1 - \mathrm{e}^{\varepsilon-(-L+\ell\Delta x)}\big)\, b_\ell^k.$$

The Plancherel Theorem gives the following:

**Lemma D.1.** *Let $\widetilde{\delta}(\varepsilon)$ and $\boldsymbol{b}^k$ be defined as above.*

*Denote $\boldsymbol{w}_\varepsilon \in \mathbb{R}^n$ such that*

$$(\boldsymbol{w}_\varepsilon)_\ell = \left[ 0, 1 - \mathrm{e}^{\varepsilon - (-L + \ell \Delta x)} \right]. \tag{D.1}$$

*Then, we have that*

$$\widetilde{\delta}(\varepsilon) = \frac{1}{n} \langle \mathcal{F}(D\boldsymbol{w}_\varepsilon), \mathcal{F}(D\boldsymbol{a}^1)^{\odot k_1} \odot \cdots \odot \mathcal{F}(D\boldsymbol{a}^m)^{\odot k_m} \rangle.$$

*Proof.* See (18). $\qquad\square$

We instantly see that if both $\mathcal{F}(D\boldsymbol{w}_\varepsilon)$ and $\mathcal{F}(D\boldsymbol{a}^i)$, $1 \leq i \leq m$, are precomputed, $\widetilde{\delta}(\varepsilon)$ can be computed in $\mathcal{O}(n)$ time (where $n$ is the number of discretisation points for the PLD).

We can utilise this by placing the individual DP hyperparameters into well-chosen buckets, and by pre-computing FFTs corresponding to the hyperparameter values of each bucket. Then, the approximative numerical PLD for each sequence of DP hyperparameters (e.g. sequence of noise ratios) can be written in a form

$$\mathcal{F}(D\boldsymbol{a}^1)^{\odot k_1} \odot \cdots \odot \mathcal{F}(D\boldsymbol{a}^m)^{\odot k_m},$$

where $k_i$'s correspond to number of elements in each bucket. If we also have $\mathcal{F}(D\boldsymbol{w}_\varepsilon)$ precomputed for different values of $\varepsilon$, we can easily construct a numerical accountant that outputs an approximation of $\varepsilon$ as a function of $\delta$.

### D.2 Noise Variance Grid for Fast Individual Accounting

We next show how to construct the DP hyperparameter grid for DP-SGD: a numerical $\sigma$-grid. We remark that Yu et al. (36) carry out similar approximations for speeding up their approximative individual RDP accountants.

Suppose we have models $a_0, \ldots, a_T$ as an output of DP-SGD iteration that we run with subsampling ratio $q$, clipping constant $C > 0$ and noise parameter $\sigma$. Also, suppose, that for a given data element $x$, along the iteration the gradients have norms

$$C_{x,i} := \|\nabla_\theta f(a_i, x)\|, \quad 0 \leq i \leq T - 1.$$

We get the individual $\varepsilon_x$-value (or individual numerical PLD, more generally) then for the entry $x$ by considering heterogeneous compositions of DP-SGD mechanisms with parameter values

$$q \quad \text{and} \quad \widetilde{\sigma}_{x,i} = \frac{C}{C_{x,i}} \cdot \sigma, \quad 0 \leq i \leq T - 1.$$

A naive approach would require up to $T$ FFT evaluations which quickly becomes computationally heavy. For the approximation, we determine a $\sigma$-grid

$$\Sigma = \{\sigma_0, \ldots, \sigma_{n_\sigma}\},$$

where $n_\sigma \in \mathbb{Z}^+$ is a number of intervals in the grid and

$$\sigma_i = \sigma_{\min} + i \cdot \frac{\sigma_{\max} - \sigma_{\min}}{n_\sigma}.$$

We then encode the sequence of noise ratios

$$\widetilde{\Sigma} := \{\widetilde{\sigma}_{x,0}, \ldots, \widetilde{\sigma}_{x,n-1}\}$$

into a tuple of integers

$$\mathbf{k} = (k_0, k_1, \ldots, k_{n_\sigma}),$$

where

$$k_i = \begin{cases} \#\{\widetilde{\sigma} \in \widetilde{\Sigma} : \sigma_i \leq \widetilde{\sigma} < \sigma_{i+1}\}, & i < n_\sigma, \\ \#\{\widetilde{\sigma} \in \widetilde{\Sigma} : \sigma_{n_\sigma} \leq \widetilde{\sigma}\}, & i = n_\sigma. \end{cases} \tag{D.2}$$

i.e. $k_i$ is number of scaled noise parameters $\widetilde{\sigma}$ hitting the bin number $i$ in the grid $\Sigma$.

By the construction of the approximation, we have the following:

**Theorem D.2.** *Consider the approximation described above. Denote the FFT transformation of the approximative numerical PLD obtained with the $\Sigma$-grid as*

$$\widetilde{a}_x = \mathcal{F}(D\widetilde{\boldsymbol{a}}^1)^{\odot k_1} \odot \cdots \odot \mathcal{F}(D\widetilde{\boldsymbol{a}}^{n_\sigma})^{\odot k_{n_\sigma}}$$

*and the corresponding $\delta$ (as a function of $\varepsilon$), as given by Lemma D.1 as*

$$\widetilde{\delta}(\varepsilon) = \frac{1}{n}\langle \mathcal{F}(D\boldsymbol{w}_\varepsilon), \widetilde{a}_x \rangle,$$

*where $\boldsymbol{w}_\varepsilon$ is the weight vector (D.1). Then, we have that for each $\varepsilon \geq 0$:*

$$\delta(\varepsilon) \leq \widetilde{\delta}(\varepsilon) + \mathrm{err},$$

*where $\delta(\varepsilon)$ is the tight value of $\delta$ corresponding to the actual sequence of noise ratios $\{\widetilde{\sigma}_{x,0}, \ldots, \widetilde{\sigma}_{x,n-1}\}$ and $\mathrm{err}$ denotes the (controllable) numerical errors arising from the discretisations of PLDs.*

*Proof.* The results follows from the data-processing inequality since each $\sigma_{x,i}$-value is placed to bucket corresponding to a smaller noise ratio. $\qquad\square$

The numerical error term $\mathrm{err}$ can also be bounded using the techniques and results of (15). The importang thing here is that if the FFTs $\mathcal{F}(D\widetilde{\boldsymbol{a}}^i)$, $0 \leq i \leq n_\sigma$, are precomputed as well as $\mathcal{F}(D\boldsymbol{w}_\varepsilon)$, then evaluating $\widetilde{\delta}(\varepsilon)$ is an $\mathcal{O}(n)$ operation.

To implement the approximative accountant described in Section 5, we numerically approximate individual upper bound $\mu$-GDP values using the bisection method.

### D.3 Poisson Subsampling of the Gaussian Mechanism

For completeness we show how to determine the PLDs for the Poisson subsampled Gaussian mechanism, required for the individual accounting of DP-SGD. Consider the Gaussian mechanism

$$\mathcal{M}(X) = \sum_{x \in X} f(x) + \mathcal{N}(0, \sigma^2 I_d),$$

where $f$ is a function $f : \mathcal{X} \to \mathbb{R}^d$. Then, if the dataset $X'$ and $X$ are neighbours such that $X = X' \cup \{x'\}$ for some entry $x'$, then from the translation invariance of the hockey-stick divergence and from the unitary invariance of the Gaussian noise, it follows that, for all $\alpha \geq 0$,

$$H_\alpha\big(\mathcal{M}_n(X)||\mathcal{M}_n(X')\big) = H_\alpha\big(\mathcal{N}\big(\|f(x')\|_2, \sigma^2\big)||\mathcal{N}\big(0, \sigma^2\big)\big).$$

Furthermore, from the scaling invariance of the hockey-stick divergence, we have that for all $\alpha \geq 0$,

$$H_\alpha\big(\mathcal{M}_n(X)||\mathcal{M}_n(X')\big) = H_\alpha\big(\mathcal{N}\big(C, \sigma^2\big)||\mathcal{N}\big(0, \sigma^2\big)\big)$$
$$= H_\alpha\big(\mathcal{N}\big(1, (\sigma/C)^2\big)||\mathcal{N}\big(0, (\sigma/C)^2\big)\big),$$

where $C = \|f(x')\|_2$. Using the subsampling amplification results of Zhu et al. (37) we get a unique worst-case pair $(P, Q)$, where

$$P = q \cdot \mathcal{N}\big(1, \widetilde{\sigma}^2\big) + (1 - q) \cdot \mathcal{N}\big(0, \widetilde{\sigma}^2\big),$$
$$Q = \mathcal{N}\big(0, \widetilde{\sigma}^2\big),$$

where $\widetilde{\sigma} = \sigma/C$. The PLD $\omega_{P/Q}$ is then determined by $P$ and $Q$ as defined in Def. 5.

## E Experiments with MIMIC-III

We use the preprocessing provided by (16) to obtain the train and test data for the phenotyping task. We refer to (16) for details on the preprocessing pipeline and the details on the phenotyping task. We tune the hyperparameters using Bayesian optimization using the hyperparameter tuning library Optuna (1) to maximise the macro-averaged AUC-ROC, the task's primary metric. We train using DP-GD and opacus (Yousefpour et al.) with noise parameter $\sigma \approx 10.61$ and determine the optimal clipping constant as $C \approx 0.79$ in our training runs. We compute the budget $B$ so that filtering starts after 50 epochs and set the maximum number of epochs to 100. With these parameter values $\varepsilon = 2.75$, when $\delta = 10^{-5}$.

### E.1 Effect of Suboptimal Hyperparameter Values on Filtered DP-GD

We study here the effect of choosing sub-optimal clipping constants by evaluating the effects of filtering using clipping constants reaching from half the optimum to five times the optimum (Figure 6). We observe that filtering only improves the utility when choosing clipping constants that are sub-optimal (e.g., 5x the optimum). Our observations complement the observations made by (12), who also observe the largest improvements by filtering in sub-optimal hyperparameter regimes.

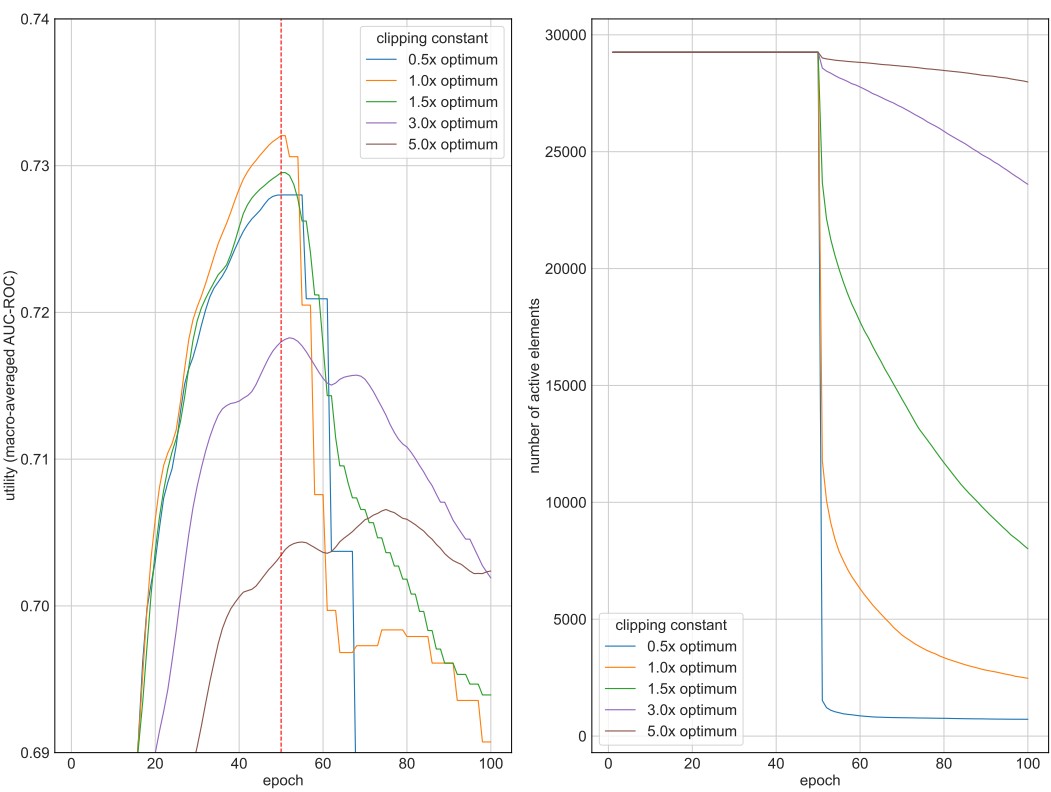

Figure 6: Filtered DP-GD with a maximum privacy loss of $(\varepsilon, \delta) = (2.75, 10^{-5})$ using different clipping constants. Rest of the hyperparameters are tuned. Left: The test AUC-ROC as a function of epochs. The red vertical line denotes the starting point of the filtering. Right: The number of active elements.

### E.2 Histograms of Individual $\varepsilon$-Values for the MIMIC-III Experiment

As described in the main text, to observe the differences across subgroups, we choose five non-overlapping groups of size 1000 based on the following criteria: subgroup 0: No diagnosis at all, subgroups 1 and 2: Pneumonia/no Pneumonia, subgroups 3 and 4: Heart attack/no heart attack. In the training data, there are total 2072 cases without a diagnosis, 4105 Pneumonia cases and in total 9413 heart attack cases. We remark that it is not uncommon for a patient to have multiple conditions.

During the training, we track the gradient norms $C_{x,i}$ for all elements of the training dataset, and thus we compute the individual $\varepsilon$-values after given number of iterations for a given value of $\delta$ ($\delta$ is set to $10^{-5}$). In Figures 7 and 8 we display histograms of the individual $\varepsilon$ values after 50 epochs. With the optimal clipping constant a majority of the datapoints have an individual $\varepsilon = 2.75$, which is near the budget. For a clipping constant that is 5x the optimum most points are significantly smaller than $\varepsilon = 2.75$.

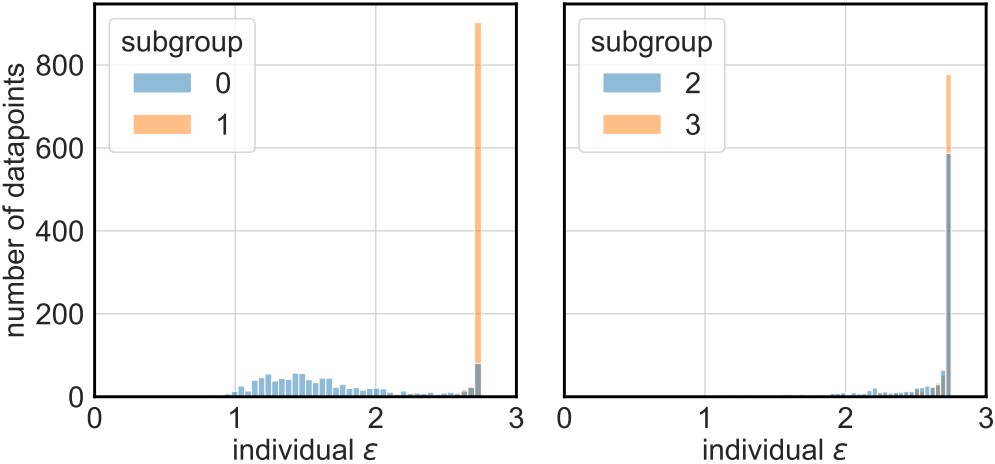

Figure 7: Histogram of individual $\varepsilon$ after 50 epochs without filtering when using the optimal clipping constant. A majority of the individual $\varepsilon$ are near $\varepsilon = 2.75$, which means that they will be deactivated in epoch 51, which uses filtering.

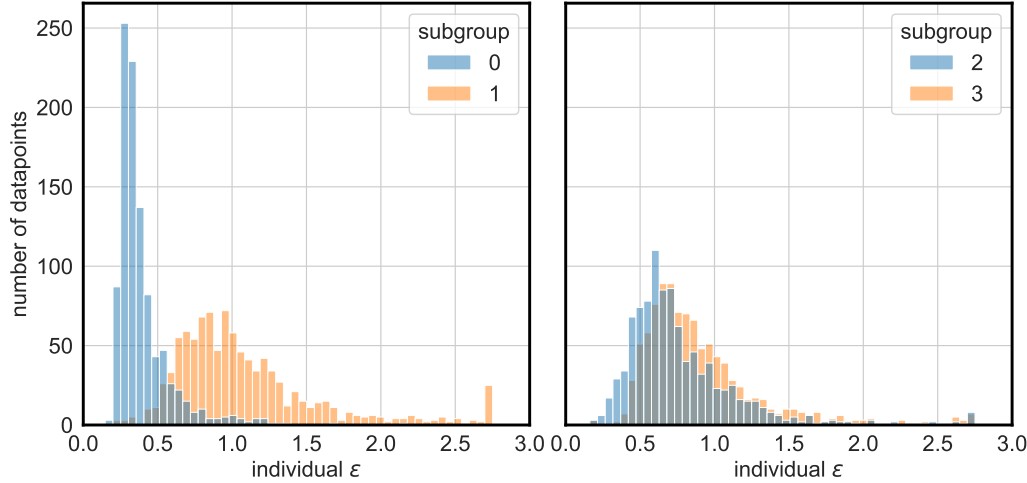

Figure 8: Histogram of individual $\varepsilon$ after 50 epochs without filtering when using a clipping constant that is 5x the optimum. A majority of the individual $\varepsilon$ are far away from $\varepsilon = 2.75$, which means that they will not be instantly deactivated in epoch 51, which uses filtering.

### E.3 FURTHER EXPERIMENTAL RESULTS

We run the same experiment as above but now, instead of maximum privacy loss of $(\varepsilon = 2.75, \delta = 10^{-5})$, using maximum privacy loss of $\varepsilon = 0.5$ and $\varepsilon = 10.0$. Figures 9 and 10 depict the performance in these cases (test AUC-ROC curves). We see, similarly to the experiments of (12), that the overall performance slightly increases when using the individual filtering.

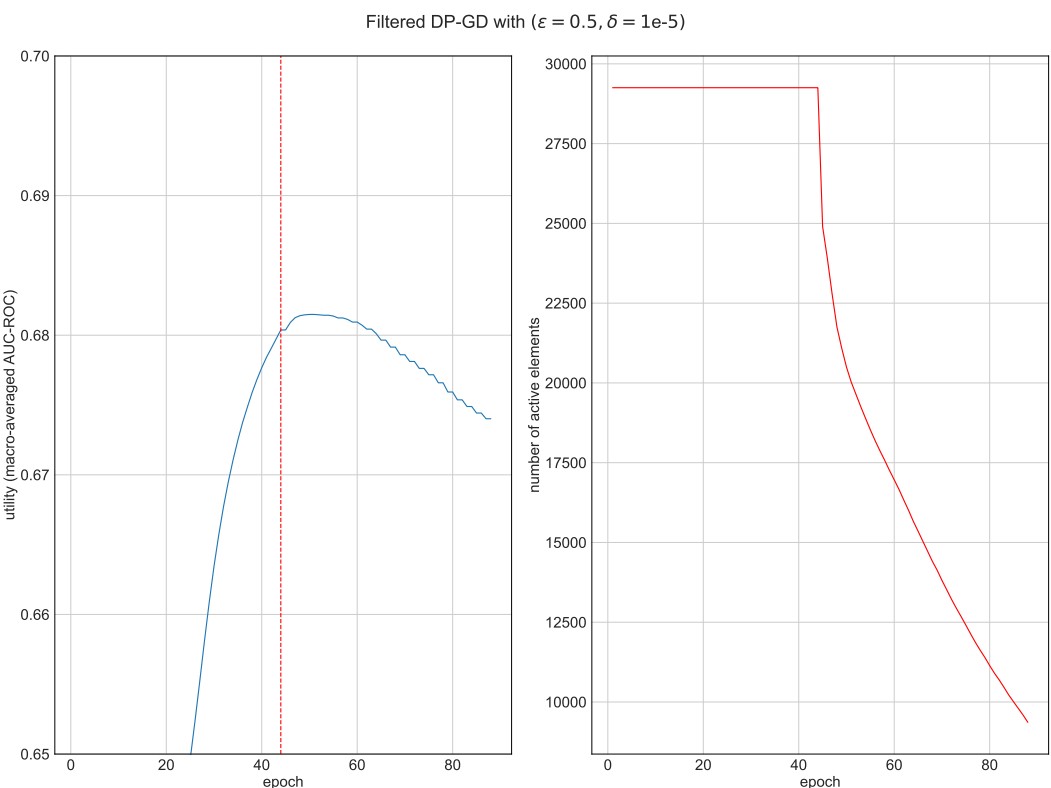

Figure 9: Filtered DP-GD with a maximum privacy loss of $(\varepsilon, \delta) = (0.5, 10^{-5})$ using tuned hyper-parameters. Left: the test AUC-ROC as a function of epochs. Right: The number of active elements.

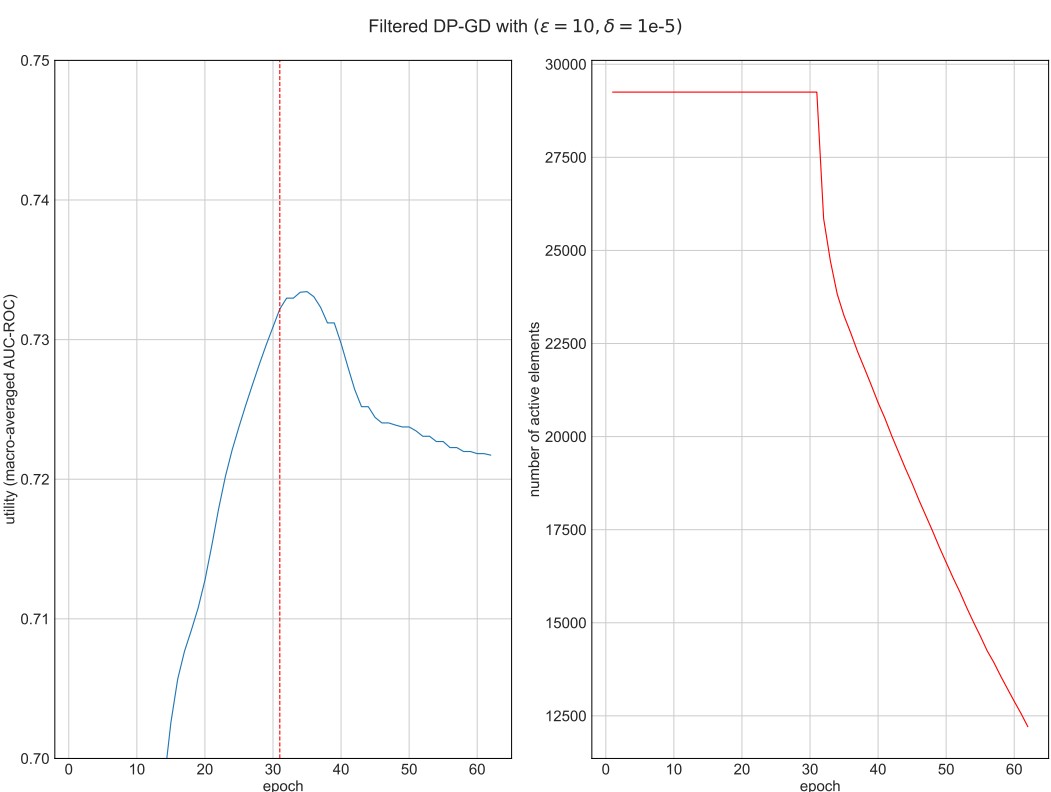

Figure 10: Filtered DP-GD with a maximum privacy loss of $(\varepsilon, \delta) = (10.0, 10^{-5})$ using tuned hyperparameters. Left: the test AUC-ROC as a function of epochs. Right: The number of active elements.

