# OpenReview forum: "Individual Privacy Accounting with Gaussian Differential Privacy"
_ICLR.cc/2023/Conference — ICLR 2023 poster_

### Official Review · Reviewer_6fJh · 2022-10-24

**Confidence:** 3
**Correctness:** 4
**Technical Novelty And Significance:** 3
**Empirical Novelty And Significance:** 3
**Recommendation:** 6

**Clarity, Quality, Novelty And Reproducibility:**

As mentioned earlier, I think the paper is well-written, so I don't have much issues with clarity, barring the questions I raised in the same section of the review.

There seems to be sufficient novelty in the techniques -- sure, many ideas do seem to be based on recent prior work, but the idea of trying GDP did the trick to improve. I'm satisfied by that!

**Strength And Weaknesses:**

Strengths:
1. The paper is well-written, with the flow and the preliminary ideas being easy to follow. The definitions and the ideas are explained quite nicely, along with supporting examples at different places. I also like the idea of keeping the appendix within the same document because for me, personally, it makes it easier to read.
2. The ideas are backed by rigorous proofs, which seem believable to me. So, the paper appears strong enough to me mathematically.
3. The experiments support the claims, too, as the improvements over the usual RDP accountants are clearly visible.
4. On a high level, I think this does seem to be an important problem to work on because of the existing gap between deployable DP systems and the theoretical foundations. Of course, more progress has been made, especially in the last few years to invent more practical algorithms for various applications, but tighter accounting is a great step here. It feels like a significant problem to study to learn about the limits of the existing DP tools or even for what we could possibly hope to achieve privately. Essentially, all I'm saying is that I like the paper also because of the problem they're trying to solve.

Weaknesses:
1. I did see comparisons with prior work, but only through experiments. I'm not *super* familiar with this particular aspect of DP literature, so I'm wondering if it were possible at all to have more in-depth theoretical comparisons, as well?
2. The experiments were somewhat convincing, but not fully, if I'm honest. For example, in Figure 1, the plot of $\delta$-vs-$\varepsilon$ doesn't give me that much details as to how it compares with the RDP accountant. Also, I'm not sure how much we gain by improving $\delta$ because any non-trivial DP algorithm usually has polylogarithmic dependence on $\delta$ in its accuracy guarantees. Isn't $\varepsilon$ the quantity we should be focussing on more? More clarity on these details would be nice, unless I'm missing something obvious.
3. I wish I could see more applications of their work besides DP-GD. That would have given me a better idea of how useful this is for other techniques that don't involve the Gaussian mechanism, for example, those that use stability-based techniques. What do I expect the results to look like in those cases, and why?

**Summary Of The Paper:**

This paper gives new methods for privacy loss accounting in differentially private (DP) algorithms, and illustrates the improvements of its methods over prior work through experimental results. The experiments are mainly performed via the case of running private gradient descent (DP-GD) in various learning tasks. As the paper lists in Section 1.1, it contains four main contributions.
1. Analysis of fully adaptive compositions of DP mechanisms using Gaussian differential privacy (GDP), which in turn gives tight $(\varepsilon, \delta)$ bounds for compositions of Gaussian mechanisms.
2. Using the idea of dominating pairs of distributions and the Blackwell Theorem, they provide an approximative $(\varepsilon, \delta)$ accountant, which frequently leads to better $\varepsilon$ bounds than the traditional Renyi differential privacy (RDP) accountants.
3. It provides efficient numerical techniques to account for individual privacy parameters using privacy loss distributions (PLD's) and and FFT, which improves on the running time of existing accountants.
4. It gives experimental results to show the improvements in privacy accounting over RDP for both the aforementioned methods. It also observes that a result of individual filtering is that it leads to disparate loss in accuracies among subgroups while training using a neural network via DP-GD.

**Summary Of The Review:**

I do vote for acceptance of this paper based on what I wrote earlier -- the positives seem to outweigh the negatives, in my opinion. That said, my review might change depending on the authors response.

---

> ### Author Response · Authors · 2022-11-18
> **response to reviewer 6fJh**
>
> > The experiments were somewhat convincing, but not fully, if I'm honest. For example, in Figure 1, the plot of $\delta$-vs-$\varepsilon$ doesn't give me that much details as to how it compares with the RDP accountant.
>
> We believe here is a slight misunderstanding: the $\delta$-vs-$\varepsilon$-curve simply depicts the accuracy of the approximative part of the method: we depict the the privacy profile for a single data element by a) computing accurately via the PLDs obtained from dominating pairs of distributions and by b) via approximative GDP upper bound that we need in order to be able to use the existing theory. The $\delta$-vs-$\varepsilon$-curve illustrates that the GDP curve is actually an accurate approximation for large number of compositions (as we expect, because of central limit theorem).
>
> The histogram on the right of Fig. 1, on the other hand, gives a comparison between RDP and our method that uses PLDs. We see that the RDP does not capture as accurately the small $\varepsilon$-values as our method.
>
> > I wish I could see more applications of their work besides DP-GD. That would have given me a better idea of how useful this is for other techniques that don't involve the Gaussian mechanism, for example, those that use stability-based techniques. What do I expect the results to look like in those cases, and why?
>
> In general, GDP gives a good approximation when the number of compositions is large (e.g., think of DP-SGD training of a deep learning model, where the number of compositions is commonly very large). This has been utilized e.g. by [1] and [2], and GDP accounting for DP-SGD is implemented also in Tensorflow Privacy and Opacus, for example. We believe our approximative method (described in Section 5, which also allows individual filtering) is useful when the number of compositions is large.
>
> > I did see comparisons with prior work, but only through experiments. I'm not super familiar with this particular aspect of DP literature, so I'm wondering if it were possible at all to have more in-depth theoretical comparisons, as well?
>
> The bigger picture here is that we aim at replicating the benefits of optimal accounting directly using the hockey-stick divergence vs. RDP accounting in case of the fully adaptive compositions analysis. It is well-known that the hockey-stick based numerical accounting often leads to much tighter bounds than the RDP accounting, see e.g. [3] and [4]. The fully adaptive analysis exists using RDP [5], and we are here able to provide the optimal accounting in case of the Gaussian mechanism (rigorously) and also an approximative method that gives accurate results for any DP mechanisms when the number of compositions is large.
>
> References:
>
> [1] Dong, Jinshuo, Aaron Roth, and Weijie J. Su. "Gaussian differential privacy." Journal of the Royal Statistical Society Series B 84.1 (2022): 3-37.
>
> [2] Bu, Z., Dong, J., Long, Q., \& Su, W. J. (2020). Deep learning with gaussian differential privacy. Harvard data science review, 2020(23).
>
> [3] Gopi, Sivakanth, Yin Tat Lee, and Lukas Wutschitz. "Numerical composition of differential privacy." Advances in Neural Information Processing Systems 34 (2021): 11631-11642.
>
> [4] Zhu, Yuqing, Jinshuo Dong, and Yu-Xiang Wang. "Optimal accounting of differential privacy via characteristic function." International Conference on Artificial Intelligence and Statistics. PMLR, 2022.
>
> [5] Feldman, Vitaly, and Tijana Zrnic. "Individual privacy accounting via a renyi filter." Advances in Neural Information Processing Systems 34 (2021): 28080-28091.

---

### Official Review · Reviewer_MJHk · 2022-10-25

**Confidence:** 4
**Correctness:** 3
**Technical Novelty And Significance:** 3
**Empirical Novelty And Significance:** 2
**Recommendation:** 6

**Clarity, Quality, Novelty And Reproducibility:**

This paper is mostly well-written. The statements in the paper are clear and concise. The novelty of this paper is not very high (see the weaknesses). I believe that the experimental results are reproducible, and I have checked the theoretical results which mostly look good to me.

**Strength And Weaknesses:**

Major strengths:
1. The Gaussian DP is a special case of f-DP which is one of the optimal DP definitions for tight privacy analysis, and the Renyi filter is also one of the most important breakthroughs for fine-grained DP analysis. It is great to see this work connect these two ideas.

Minor strengths:
1. The paper is well-written and easy to read.

Major weaknesses:
1. The results in this paper are not as comprehensive and novel as I expected. It is natural to use the idea of dominating pair of distributions which is equivalent to f-DP. Since Gaussian DP is mostly useful in asymptotic composition results, and the tight privacy analysis often requires f-DP, I would like to know why the proof of Theorem 11 cannot be generalized to f-DP.
2. The Blackwell theorem used in Section 5 is not very appropriate since it is based on the relationship between two tradeoff functions (as shown in the Gaussian DP paper). There is no such result in the dominating pair paper (Zhu et al. 2022). I mean, the authors should provide or cite such a result that if $(P,Q)$ is a dominating pair, then the tradeoff function $T_{P,Q}$ is a lower bound for the tradeoff functions of neighboring datasets. By the way, I guess the stochastic transformation $T$ is a Markov kernel instead of a Markov process, therefore there is a typo on Page 8.
3. The experimental results are not very related to the main topic of this paper.

Minor weaknesses:
1. The composition result of Gaussian DP is not stated in the main text (the proof description of Theorem 11) but it must be used somewhere (I know it is shown at the end of Appendix Section B, but I cannot find it in the main text.). This confuses me since in the proof of Theorem 12, the authors used $\sum_{i=1}^t \mu_i \geq B$ instead of $\sum_{i=1}^t \mu_i^2 \geq B^2$ (which I think is a typo and appears again in the line above Equation (5.1).)
2. Algorithm 1 may need to be revised. I suggest the authors check 'Algorithm 2 by Feldman and Zrnic (2021)' again since we need $(X, X^{-i})$ to be any $(S_1,S_2) \in \mathcal{S}(X_i,n)$. By the way, I think $\mu_{j+1}^{(i)}$ should be $\mu_{j}^{(i)}$
3. Theorem 12 should be put before Algorithm 1 since the two conditions, HALT and CONT, need to be defined.
4. In Section 4.1, the worse-case analysis is said to be $(\varepsilon=1.0, \delta=10^{-5})$, but this is not shown in Figure 3. Is Figure 3 an average result of individual privacy accounting?
5. In Section 6, the authors claim that 'we find that while the individual filtering leads to more equal group-wise $\varepsilon$ values, ...' I cannot find any result that shows the 'more' equal values. There seems to be no such comparison in Figure 2.


**Summary Of The Paper:**

This paper generalizes the Renyi Filter techniques and individual privacy accounting results by Feldman and Zrnic (2021) from Renyi DP to Gaussian DP. At the end of this paper, the authors use experiments with MIMIC-III to obtain a similar result to Yu et al. (2022) which shows a correlation between the average of individual privacy guarantees and the accuracies among different groups.

**Summary Of The Review:**

This paper is easy to read and follow. The authors connect the existing work together (Gaussian DP, dominating pairs, and Renyi Filter), so their results are natural and correct. It is like an exact follow-up to the work done by Feldman and Zrnic (2021). There are minor issues in this paper but I believe all are fixable.

---

> ### Author Response · Authors · 2022-11-17
> **answers to reviewer MjHk**
>
> > It is natural to use the idea of dominating pair of distributions which is equivalent to f-DP. Since Gaussian DP is mostly useful in asymptotic composition results, and the tight privacy analysis often requires f-DP, I would like to know why the proof of Theorem 11 cannot be generalized to f-DP.
>
> Let us explain the technical difficulty. To show the submartingale property for the fully adaptive composition using the hockey-stick divergence (or using $f$-DP), we have to start bounding from the last mechanism one-by-one using the dominating pairs of distributions. Once we have bounded the last mechanism in a sequence of $n$ mechanisms, the outputs of first $n-1$ are random. The difficulty is that the dominating pair of distributions for the last mechanism is determined by the output of the first $n-1$ mechanisms. Thus we cannot bound the $(n-1)$st mechanism with it's dominating pairs of distributions ( we cannot integrate it out), unless we have a dominating pairs of distributions for the last mechanism that depends only on the outputs of $n-2$ first mechanisms. The dominating pairs of distributions for the $(n-1)$st mechanism is determined by the outputs of $n-2$ first mechanisms, and if we could find a pair of distributions that both dominates the last mechanism and 'fills' the privacy profile
> such that the resulting non-adaptive composition would be tightly dominated by some pairs of distribution (or tightly $f$-DP for some $f$), and we could repeat this mechanism by mechanism, and we believe then this would be possible. For GDP this is fairly straighforward and this is what we do. So, fundamentally, we believe the difficulty follows from the fact that the Blackwell theorem does not  give a decomposable (factorizable) Markov kernel as such.
>
> > The Blackwell theorem used in Section 5 is not very appropriate since it is based on the relationship between two tradeoff functions (as shown in the Gaussian DP paper). There is no such result in the dominating pair paper (Zhu et al. 2022). I mean, the authors should provide or cite such a result that if  is a dominating pair, then the tradeoff function  is a lower bound for the tradeoff functions of neighboring datasets. By the way, I guess the stochastic transformation  is a Markov kernel instead of a Markov process, therefore there is a typo on Page 8.
>
>
> There is in fact one-to-one relationship between the ordering determined by trade-off functions and the hockey-stick divergence and this follows from the results by Zhu et al. [1]. Let $P$ and $Q$ be probability distributions and denote by $T(\widetilde{P},\widetilde{Q})$ the trade-off function determined by $P$ and $Q$ and let $H_\alpha(\widetilde{P}||\widetilde{Q})$ denote the hockey-stick divergence.
> The Lemma 20 of Zhu et al. [1] is a restatement of
>  Proposition 2.12 by Dong et al. [1] (Prop. 2.12 of the Arxiv version) and it states that for any $\varepsilon\in \mathbb{R}$,
>
> $ H_{e^{\varepsilon}}(P||Q) = 1 + T[P,Q]^*(-e^{\varepsilon}),$
>
> where, for a trade-off function $f$, $f^*$ denotes
>
> $
> f^*(y) = \sup_{x \in \mathbb{R}} yx - f(x).
> $
>
> From this is the equivalence follows directly, i.e.
>
> $
> H_{e^{\varepsilon}}(\widetilde{P}||\widetilde{Q}) \leq H_{e^{\varepsilon}}(P||Q)
> $
>
> for all $\varepsilon\in \mathbb{R}$,
> if and only if
>
> $
> T[\widetilde{P},\widetilde{Q}] \geq T[P,Q].
> $
>
> We have added this clarification also the revised
> version of the paper.
>
> >Theorem 12 should be put before Algorithm 1 since the two conditions, HALT and CONT, need to be defined.
>
> Thank you pointing this out, we have made corresponding change to the main text.
>
> > Algorithm 1 may need to be revised. I suggest the authors check 'Algorithm 2 by Feldman and Zrnic (2021)' again.
>
> Indeed, thank you for pointing this out. Our version was valid only for linear queries. We have changed this accordingly.
>
> > In Section 4.1, the worse-case analysis is said to be $(\varepsilon=1.0,\delta=10^{-5})$-DP.
>
> Thank you for pointing out, it is indeed $(\varepsilon=0.8,\delta=10^{-5})$-DP. This has been corrected in
> the revised version.
>
> References:
>
> [1] Zhu, Yuqing, Jinshuo Dong, and Yu-Xiang Wang. "Optimal accounting of differential privacy via characteristic function." International Conference on Artificial Intelligence and Statistics. PMLR, 2022.
>
> [2] Dong, Jinshuo, Aaron Roth, and Weijie J. Su. "Gaussian differential privacy." Journal of the Royal Statistical Society Series B 84.1 (2022): 3-37.

---

> > ### Comment · Reviewer_MJHk · 2022-11-17
> > **Thanks for your response!**
> >
> > 1. Thanks for the explanation. It sounds reasonable to me.
> >
> > 2. This argument sounds good to me. Please consider adding this to your main text or appendix. By the way, currently, it seems that Section 5 is still the same as the original version. Please update it.
> >
> > 3. It looks good to me now.
> >
> > 4. It looks good to me now, except that the data element should be $X^{i}$ instead of $X^{-i}$, right?
> >
> > 5. It looks good to me now.
> >
> > I would raise my recommendation score after I see the update of Section 5.

---

> > > ### Author Response · Authors · 2022-11-18
> > > **response to reviewer MjHk**
> > >
> > > Thank you for the comments! We have addressed these. We have added the argument relating dominating pairs and the Blackwell theorem (in a polished form) to the Appendix section C.6 and we refer to it from Section 5 which we have also updated now.

---

### Official Review · Reviewer_ERD6 · 2022-10-26

**Confidence:** 3
**Correctness:** 4
**Technical Novelty And Significance:** 2
**Empirical Novelty And Significance:** 2
**Recommendation:** 5

**Clarity, Quality, Novelty And Reproducibility:**

The paper is technically solid. However, the presentation can be improved to highlight the main results. The techniques used are mainly from existing works.

**Strength And Weaknesses:**

Strengths
- The proposed accountant method achieves tighter bounds for the Gaussian mechanism than the commonly used RDP accounting.
- Efficient numerical calculation method is proposed.

Weakness
- The writing can be improved. Overall the paper looks very technical. The main results and their implications do not stand out well.
- Technically this work mainly uses existing techniques from previous works.
- More experiments can help to sufficiently demonstrate the effectiveness of the proposed PLD method
- The disparate loss among subgroups for DP is observed in previous works and this part is somewhat orthogonal to the main contribution of this work.

**Summary Of The Paper:**

This work derives privacy accounting for Gaussian-DP under fully adaptive composition, which gives tighter privacy bounds than RDP analysis. Efficient numerical techniques are proposed to compute the privacy parameters in practice. The theoretical findings are verified by experiments.

**Summary Of The Review:**

This work derives tighter privacy accounting for Gaussian mechanisms with Gaussian-DP than RDP accounting. However the presentation needs improvement, and technical contributions is somewhat limited.

---

> ### Author Response · Authors · 2022-11-18
> **response to reviewer ERD6**
>
> > Technically this work mainly uses existing techniques from previous works.
>
> We believe that the proof technique for GDP analysis of fully adaptive compositions does require some non-trivial extensions to the existing theory. Our proof uses the dominating pairs of distributions [1] and a suitable hockey-stick divergence - based supermartingale that we propose. The fast numerical method we propose for evaluating the individual $\varepsilon$-values using privacy loss distributions (PLDs) is novel (we are able to compute the individual $\varepsilon$'s with $\mathcal{O}(n)$ computational cost, where $n$ is the number of discretization points for the PLD, instead of $\mathcal{O}(n \log n)$ cost that would be obtained by a naive application of FFT) .
>
> > The disparate loss among subgroups for DP is observed in previous works and this part is somewhat orthogonal to the main contribution of this work.
>
> We agree it is observed for DP in general in the previous works, but the fact that filtering further amplifies this effect has not been addressed. Although filtering may slightly increase the overall accuracy (see e.g. [2] and the experimental results we have added in Appendix E.3 of the revised manuscript), it comes at the cost of even more unequal model accuracies between subgroups. We believe this kind of holistic view, when possible, is important when considering different ethical aspects of ML algorithms. We remark that filtering is only one application of the fully adaptive analysis and therefore does not decrease its importance (see e.g. [3] for applications of fully adaptive analysis in DP deep learning).
>
> > The writing can be improved. Overall the paper looks very technical. The main results and their implications do not stand out well.
>
> We have tried to improve the presentation in the revised version, especially in Section 5 (see also comments of reviewer MjHk).
>
> References:
>
> [1] Zhu, Yuqing, Jinshuo Dong, and Yu-Xiang Wang. "Optimal accounting of differential privacy via characteristic function." International Conference on Artificial Intelligence and Statistics. PMLR, 2022.
>
> [2] Feldman, Vitaly, and Tijana Zrnic. "Individual privacy accounting via a renyi filter." Advances in Neural Information Processing Systems 34 (2021): 28080-28091.
>
> [3] L\'ecuyer, Mathias. "Practical Privacy Filters and Odometers with R\'enyi Differential Privacy and Applications to Differentially Private Deep Learning." arXiv preprint arXiv:2103.01379 (2021).

---

### Official Review · Reviewer_wyyZ · 2022-10-31

**Confidence:** 3
**Correctness:** 4
**Technical Novelty And Significance:** 3
**Empirical Novelty And Significance:** 2
**Recommendation:** 6

**Clarity, Quality, Novelty And Reproducibility:**

The paper is well-written and clearly positions itself within the relevant literature. Code for the experiments is provided in the supplement.

**Strength And Weaknesses:**

## Strengths
- The paper has adequate review of the relevant literature, is mostly self-contained, and clearly states the prior results it builds upon.
- It provides a proof for a natural (and perhaps not surprising) extension of the prior individual privacy accounting in the RDP case.

## Weaknesses
- The technical contribution seems incremental (proof of Theorem 11), but I will leave the assessment of the paper's novelty to the experts in this particular topic.
- Experiments should better highlight the practical use of individual GDP accounting compared to RDP. Figure C.3 shows one example for private GD, with a fixed number of steps. Perhaps showing similar plots under different settings (e.g. different number of steps) can help.
- Experiments in Section 6 merit further discussion. Besides the disparate impact problem, the experiments suggest that even in terms of overall AUC (Figure 4), very soon after filtering starts, the overall AUC begins to drop, which suggest a limited benefit from individual accounting. Did the authors compare to not using individual accounting?

**Summary Of The Paper:**

The paper extends previous results on individual privacy accounting from Rényi DP to Gaussian DP. It also considers methods to maintain approximate privacy filters.

**Summary Of The Review:**

A nice extension of previous individual accounting results, from the Rényi DP to the Gaussian DP case.

---

> ### Author Response · Authors · 2022-11-18
> **response to reviewer wyyZ**
>
>
> > The technical contribution seems incremental (proof of Theorem 11)
>
> We believe that the proof technique for GDP analysis of fully adaptive compositions does require some non-trivial extensions to the existing theory. Our proof uses the dominating pairs of distributions [3] and a suitable hockey-stick divergence - based supermartingale that we propose. The fast numerical method we propose for evaluating the individual $\varepsilon$-values using privacy loss distributions (PLDs) for DP-SGD is novel (we are able to compute the individual $\varepsilon$'s with $\mathcal{O}(n)$ computational cost, where $n$ is the number of discretization points for the PLD, instead of $\mathcal{O}(n \log n)$ cost that would be obtained by a naive application of FFT) .
>
> > Experiments should better highlight the practical use of individual GDP accounting compared to RDP. Figure C.3 shows one example for private GD, with a fixed number of steps. Perhaps showing similar plots under different settings (e.g. different number of steps) can help.
>
> We have added two new plots in Appendix Section C.4 to illustrate the effect of number of steps: $(\varepsilon,\delta)$-curves for different numbers of steps and also a plot of max number of steps vs. privacy budget $\varepsilon$ (for fixed $\delta$).
>
> > Experiments in Section 6 merit further discussion. Besides the disparate impact problem, the experiments suggest that even in terms of overall AUC (Figure 4), very soon after filtering starts, the overall AUC begins to drop, which suggest a limited benefit from individual accounting. Did the authors compare to not using individual accounting?
>
> We agree that the gain in AUC improvement when using (nearly) optimal hyperparameters for training is small (see also the experimental results we have added in Appendix E.3 of the revised manuscript). However, the gain can be larger when training with suboptimal hyperparameters, e.g., in federated settings where the optimal hyperparameters are specific to the client (see also discussion of Feldman and Zrnic [1]). The AUC of the model can be queried repeatedly and filtering can be stopped once the AUC starts to drop and thus we believe that filtering allows for training additional steps for as many as improve the model. We remark that filtering is only one application of the fully adaptive analysis and therefore does not decrease its importance (see e.g. [2] for applications of fully adaptive analysis in DP deep learning).
>
> References:
>
> [1] Feldman, Vitaly, and Tijana Zrnic. "Individual privacy accounting via a renyi filter." Advances in Neural Information Processing Systems 34 (2021): 28080-28091.
>
> [2] L\'ecuyer, Mathias. "Practical Privacy Filters and Odometers with R\'enyi Differential Privacy and Applications to Differentially Private Deep Learning." arXiv preprint arXiv:2103.01379 (2021).
>
> [3] Zhu, Yuqing, Jinshuo Dong, and Yu-Xiang Wang. "Optimal accounting of differential privacy via characteristic function." International Conference on Artificial Intelligence and Statistics. PMLR, 2022.

---

### Comment · Area_Chair_P5Ba · 2022-12-13
**Question about the quantitative improvement**

If I understand correctly the main benefit of this work is the quantitative improvement one would get from using GDP over the original Renyi filter of Feldman and Zrnic applied to the Renyi DP parameters of a Gaussian (since, when applied to the composition of Gaussians their result will give the same Renyi bounds as those of a Gaussian for all orders) . The improvement stated in this work is about 20% but does not explicitly specify how the approximate DP parameters were derived. Note that the conversion method used in TF.privacy is not static and was updated over the past year. Recent results, given in Figure 1
https://arxiv.org/abs/2004.00010 show that with better conversion of Renyi to approximate DP the gap is typically 5% or so. A better, but somewhat harder to implement formula was also given in https://arxiv.org/abs/2001.05990
Could you please clarify whether the comparison was based on up-to-date conversion techniques? If not please comment on how that would affect the results.

---

### Decision · Program_Chairs · 2023-01-20

**Decision:**

Accept: poster

**Justification For Why Not Higher Score:**

Technical novelty is limited.

**Justification For Why Not Lower Score:**

Clean theoretical result that is also likely to be useful practically

**Metareview: Summary, Strengths And Weaknesses:**

This work extends fully adaptive differential privacy filter and individual privacy techniques of Feldman and Zrnic (2019) to Gaussian DP. It demonstrates a numerical improvement in the parameters that can be achieved through this generalization and a faster way to perform individual DP accounting via existing techniques. DP composition and its individual version are one of the key tools in private ML. While the novelty of the techniques needed for this extension is relatively modest this tool is likely to be useful in ML applications.

**Note From Pc:**

if the above contains the word "oral" or "spotlight" please see: "oral" presentation means -> notable-top-5% and "spotlight" means -> notable-top-25%. As stated in our emails, we are disassociating presentation type from AC recommendations

**Summary Of Ac-Reviewer Meeting:**

No meeting but I also read the paper myself